# Arabidopsis MYB47 and MYB95 transcription factors regulate jasmonate-inducible ER-body formation
Jakub Bizan[1,6], Shayan Sarkar[1,4,6], Arpan Kumar Basak[1,5], Subhankar Bera[1], Shino Goto-Yamada [1],
Kaichiro Endo[1], Katarzyna Tarnawska-Glatt[1], Rituraj Batth[1], Kritika Bhardwaj[1],
Mohamadreza Mirzaei [1,2], Paweł Czerniawski[3], Paweł Bednarek [3] & Kenji Yamada [1] ✉

Endoplasmic reticulum (ER)-derived subcellular structures, namely ER bodies, are involved in glucosinolate-based chemical defense against insect pests and pathogenic fungi in the Brassicaceae family. In *Arabidopsis thaliana*, treatment of rosette leaves with the wounding hormone jasmonate (JA) induces *β-GLUCOSIDASE 18* (*BGLU18*) and *TSK-ASSOCIATING PROTEIN 1* (*TSA1*) gene expression, whose products accumulate in JA-inducible ER bodies; however, the underlying transcriptional regulatory mechanisms remain unknown. Here, we show that two paralogous Arabidopsis MYBs, namely MYB47 and MYB95, regulate *TSA1* and *BGLU18* expression and ER-body formation in response to JA. MYB47 and MYB95 bind to and activate the *TSA1* promoter. *TSA1* and *BGLU18* expression levels are reduced in the JA-treated rosette leaves of *myb47,95* mutants, suggesting that these MYBs play a key role in the activation of these genes. Transcriptome analysis reveals that MYB47 and MYB95 regulate a subset of JA-responsive genes, including ER-body and defense-related genes. Phylogenetic analysis shows that MYB47 and MYB95 belong to a MYB subfamily unique to the Brassicales order. Together, our findings indicate that MYB47 and MYB95 have evolved to regulate unique downstream target genes in response to JA, which include JA-inducible ER body genes important for protecting plants from fungal and herbivore attacks in Brassicaceae.

Brassicaceae plants and their close relatives have developed unique endoplasmic reticulum (ER)-derived structures, namely ER bodies (also known as dilated cisternae or fusiform bodies)[1–3]. ER bodies are subdomains of the ER that accumulate particular β-glucosidases (BGLUs), which possess an ER retention signal[3,4]. In *Arabidopsis thaliana*, almost all epidermal cells in seedlings and roots accumulate ER bodies that store vast amounts of BGLU23 (also known as PYK10) protein[3]. BGLU23 is involved in chemical defense against herbivores and fungi by providing defensive compounds from glucosinolates, which are produced uniquely in Brassicaceae[5–8]. In Arabidopsis seedlings and roots, a basic helix-loop-helix (bHLH) transcription factor NAI1 (also known as bHLH020) regulates the biogenesis of ER bodies[9]. An ER body protein, NAI2, is responsible for the formation of ER bodies in seedlings and roots by interacting with BGLU23[5,10]. NAI1

controls a specific set of genes, such as *NAI2* and *BGLU23*, involved in the formation and function of ER bodies[9,11], together with genes encoding MEMBRANE PROTEIN OF ER-BODY 1 (MEB1), MEB2, JACALIN-RELATED LECTINS (JALs), and GDSL-LIKE LIPASES (GLLs)[12–15]. JALs and GLLs form a protein complex with BGLU23 and modulate BGLU activity in the homogenate of Arabidopsis seedlings[13,14].

Rosette leaves accumulate ER bodies only in large pavement cells within the epidermis[7,16,17]. However, wounding of rosette leaves triggers the de novo formation of ER bodies in the whole leaf epidermis through the perception of the wounding hormone jasmonate (JA) after leaf-wounding[3,18]. The formation of JA-inducible ER bodies involves the expression of *TSA1* and *BGLU18*, the homologs of *NAI2* and *BGLU23*, respectively[19–21]. The Arabidopsis *nai1* mutant accumulates ER bodies in

[1]Malopolska Centre of Biotechnology, Jagiellonian University, Krakow, Poland. [2]Doctoral School of Exact and Natural Sciences, Jagiellonian University, Krakow, Poland. [3]Institute of Bioorganic Chemistry, Polish Academy of Sciences, Poznan, Poland. [4]Present address: Pediatrics Nutrition, Children's Nutrition Research, Baylor College of Medicine, Houston, Texas, USA. [5]Present address: Department of Plant-Microbe Interactions, Max-Planck Institute for Plant Breeding Research, Cologne, Germany. [6]These authors contributed equally: Jakub Bizan, Shayan Sarkar. ✉e-mail: kenji.yamada@uj.edu.pl

rosette leaves but shows no changes in *TSA1* expression when treated with JA[9,20], suggesting that NAI1 does not regulate *TSA1* expression to induce ER-body formation. On the contrary, neither the accumulation of JA-inducible ER bodies nor the expression of *TSA1* and *BGLU18* were detected in the JA insensitive mutants, *coi1-1* and the *myc2 myc3 myc4* (*myc2,3,4*) triple mutant[16,18,20]. Therefore, distinct regulatory mechanisms seem to be involved in the formation of JA-inducible ER bodies in rosette leaves and constitutively existing ER bodies in seedlings and roots.

MYB transcription factors comprise a large family of proteins that harbor a conserved DNA-binding domain and function in various biological processes[22–25]. Most land plant MYB transcription factors are separated into subfamilies conserved across various species[24,26,27]. However, some MYB transcription factors are unique to specific plant lineages and form lineage-specific subfamilies. In Arabidopsis, three such lineage-specific MYB subfamilies can be identified, which are MYB28/29/76, MYB34/51/122, and MYB47/95 subfamilies[26]. MYB28, MYB29, and MYB76 are involved in the regulation of aliphatic glucosinolate biosynthesis genes[28–30], while MYB34, MYB51, and MYB122 are involved in the regulation of indolic glucosinolate biosynthesis genes[31–33]. The biosynthesis of these glucosinolates is unique to the plant lineage including Brassicaceae. Therefore, MYB28/29/76 and MYB34/51/122 subfamilies seem to have evolved uniquely for the biosynthesis of glucosinolates, a group of defense-related chemicals in Brassicaceae. MYB47 has been shown to be involved in seed longevity, drought responses, and senescence[34–36], suggesting that this subfamily functions during a stress response. However, the available information about the function of the MYB47/95 subfamily is limited.

The involvement of the COI1 JA receptor and MYC2, MYC3, MYC4 (MYC2/3/4) transcription factors in the ER body formation highlights crucial roles of the JA signaling pathway in wound-inducible ER body formation[16,18,20]. Since MYC2/3/4 are the master transcriptional regulators of the JA-signaling pathway[37–39], specific transcription factors may exist to regulate a subset of JA-inducible ER body-related genes. Here, we report that MYB47 and MYB95 regulate *TSA1* and *BGLU18* expression and JA-inducible ER body formation in Arabidopsis. Both MYB47 and MYB95 recognized a DNA motif in the *TSA1* promoter in vitro and activated *TSA1* gene promoter in vivo. *MYB47* and *MYB95* also regulated a subset of JA-responsive genes. *MYB47* and *MYB95* expression levels were increased by JA treatment in a MYC2/3/4-dependent manner. The expression of *MYB95* was higher than that of *MYB47*, indicating that MYB95 plays a more important role in the JA-inducible ER body formation and *TSA1* and *BGLU18* expression. BGLU18 accumulation and ER body formation were reduced in the *myb95* single and *myb47,95* double mutant. However, the glucosinolate biosynthesis and seedling ER body formation were not affected in the *myb47,95* double mutant. Collectively, our findings suggest that a uniquely evolved MYB47 and MYB95 control a subset of JA response, including wound- and JA-inducible ER-body formation.

## Results
### MYB47 and MYB95 Represent Potential Regulators of *BGLU18* and *TSA1* Genes
Previous studies revealed that *TSA1* and *BGLU18* gene expression levels are upregulated after wounding and JA treatment, and they are involved in the ER-body formation in wounded and JA-treated leaves[19–21]. MYC2/3/4 are responsible for the expression of *TSA1* and *BGLU18*[20]. Since MYC2/3/4 can upregulate downstream transcription factors, it is possible that MYC2/3/4 indirectly upregulate *TSA1* and *BGLU18*. Based on the fact that transcription factors and their target gene expressions are frequently correlated[15], we screened for transcription factor-encoding genes whose expression pattern is consistent with *TSA1* and *BGLU18* expression in the gene co-expression database[40]. We found that the expression patterns of the two paralogous *MYB* genes, *MYB47* (At1g18710) and *MYB95* (At1g74430), showed a high correlation with those of *TSA1* and *BGLU18* (Table 1, Supplementary Tables 1 and 2).

Interestingly, genome-wide association study (GWAS) using publicly available gene expression data from Arabidopsis leaves[41] revealed that single

**Table 1 | Expression correlation of *BGLU18*, *TSA1*, *MYB47* and *MYB95***

| Gene | Locus | LS (BGLU18) | LS (TSA1) | LS (MYB47) | LS (MYB95) |
|---|---|---|---|---|---|
| *BGLU18* | At1g52400 | 14.2 (-) | 14.2 (1) | 5.4 (10) | 5.5 (26) |
| *TSA1* | At1g52410 | 14.2 (1) | 14.2 (-) | 5.2 (12) | 6.0 (12) |
| *MYB47* | At1g18710 | 5.4 (36) | 5.2 (46) | 14.2 (-) | 7.1 (4) |
| *MYB95* | At1g74430 | 5.5 (30) | 6.0 (30) | 7.1 (3) | 14.2 (-) |

LS, Logit score in ATTEDII ver10.1. Rank orders for each gene in the LS are shown in parentheses.

nucleotide polymorphisms (SNPs) in the *MYB47* gene are associated with variations in the *TSA1* and *BGLU18* expression levels along with *MYB47* expression level among various Arabidopsis accessions (Supplementary Fig. 1). The finding may suggest that SNPs on *MYB47* are correlated with *TSA1* and *BGLU18* gene expression in leaves of certain Arabidopsis accessions without exogenous JA treatment but presumably endogenous JA-dependent *BGLU18* expression[7,16]. Altogether, these results suggest that MYB47 and MYB95 are candidate transcription factors for regulating the expression of *TSA1* and *BGLU18*.

### JA-inducible Expression of *BGLU18* and *TSA1* Is Reduced in *myb95* and *myb47 myb95* Mutants
*MYB47* and *MYB95* are paralogous genes to each other as they are located within a duplicated region on chromosome 1[42]. Their expression patterns are correlated with each other (Table 1, Supplementary Tables S3 and S4), suggesting that these two genes likely perform similar functions. To investigate whether both MYB proteins act redundantly to control *TSA1* and *BGLU18* genes in the exogenous JA treatment, we examined *TSA1* and *BGLU18* expression and the accumulation of the encoded proteins in T-DNA insertion mutants of *MYB47* and *MYB95* genes (Fig. 1). *myb47* was a knockdown mutant whereas *myb95* was a knockout mutant (Supplementary Fig. 2). Quantitative real-time PCR (qRT-PCR) and immunoblot analyses revealed increased transcript and protein accumulation of *TSA1* and *BGLU18* in the JA-treated wild-type leaves. The JA-treated *myb95* single mutant leaves showed a clear reduction in JA-induced *TSA1* and *BGLU18* transcript levels compared with the JA-treated wild-type, but the *myb47* single mutant did not show any reduction of these genes compared to the wild-type (Fig. 1a). The expression of *TSA1* and *BGLU18* in the JA-treated leaves of the *myb47,95* double mutant was further reduced compared with the *myb95* single mutant, and immunoblot analysis showed a strongly reduced accumulation of BGLU18 and TSA1 proteins in the *myb47,95* double mutant compared with wild-type plants (Fig. 1b). These findings suggest a predominant role of MYB95 and a supportive role of MYB47 in the activation of *TSA1* and *BGLU18* genes in response to JA.

### Reduction in JA-inducible ER-body Accumulation in *myb95* and *myb47 myb95* Mutants
Since *TSA1* and *BGLU18* are involved in the ER body formation[19–21], and their expression levels are lower in the *myb95* single and *myb47,95* double mutants, we hypothesized that the ER body formation is reduced in these mutants. We examined ER-body formation in the leaves of *myb47*, *myb95*, and *myb47,95* mutants in response to JA. The results showed a clear reduction in the number of JA-inducible ER bodies in the *myb95* single and *myb47,95* double mutants compared with wild-type plants (Fig. 2). The inducibility of ER-body formation in the *myb47* single mutant was the same as that in the wild-type. However, when comparing *myb95* and *myb47,95* double mutants, the ER-body formation tends to be reduced in the *myb47,95* double mutant. Thus, the results indicate the predominant role of MYB95 and the supportive role of MYB47 in ER-body formation in response to JA, similarly to *TSA1* and *BGLU18* gene induction.

ER body formation is induced in the epidermal cells surrounding the wounded sites in leaves through a COI1-dependent manner[16,18]. Therefore,

**Fig. 1 | MYB47 and MYB95 are required for the expression of *BGLU18* and *TSA1* genes.**
**a** Induction of the *BGLU18* gene in Arabidopsis leaves. Rosette leaves from 14-d-old aseptically-grown wild-type (WT), *myb47* and *myb95* single mutants, and *myb47,95* double mutant were cut and floated on distilled water (mock) or 50 μM methyl jasmonate (JA) for 2 days and then subjected to quantitative real-time PCR (qRT-PCR) analysis. Error bars represent standard error (SE; $n = 6$ independent experiments). Different lowercase letters indicate significant differences ($p < 0.05$; Tukey's test). **b** Induction of the *TSA1* gene in Arabidopsis leaves. Rosette leaves were treated as described in (**a**). Error bars represent SE ($n = 6$ independent experiments). Different lowercase letters indicate significant differences ($p < 0.05$; Tukey's test). **c** Accumulation of BGLU18 and TSA1 proteins in Arabidopsis leaves. Rosette leaves were treated as described in (**a**), and the extracted proteins were subjected to immunoblot analysis with anti-BGLU18 antibody or anti-NAI2 antibody (cross-react with TSA1). The bands corresponding to full-length (upper) and partially degraded (lower) TSA1 proteins are shown with arrowheads. Coomassie brilliant blue (CBB) staining shows the RuBisCo large subunit (RbcL) used as a loading control.

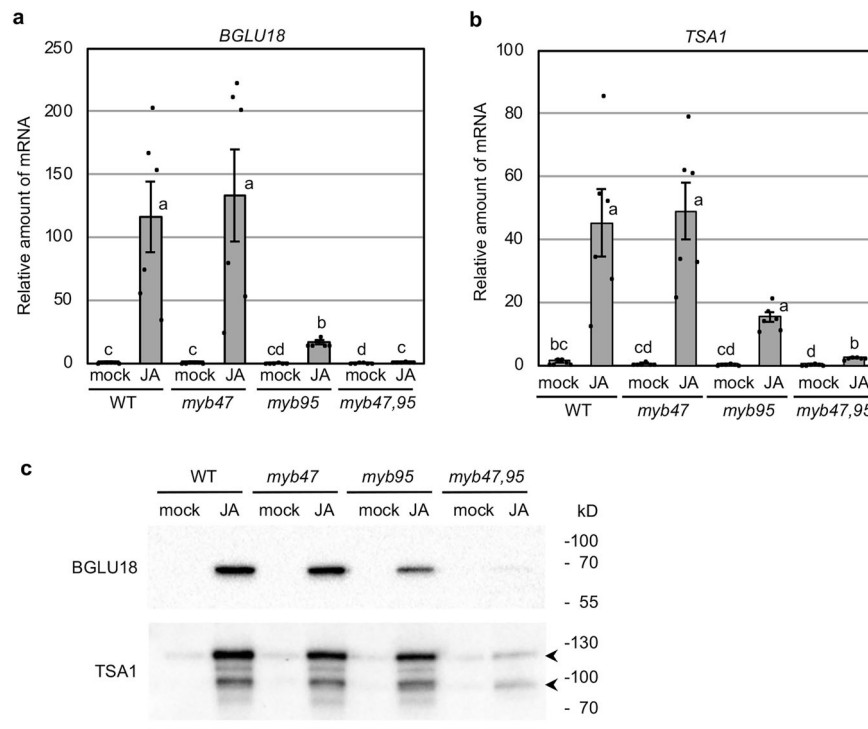

we performed a wounding experiment in *myb47,95* double mutant leaves to examine whether ER body formation occurs. The results show that the *myb47,95* double mutant reduces the accumulation of wound-inducible ER bodies compared with the wild-type (Supplementary Fig. 3), indicating that these MYBs contribute to wound-inducible ER body formation.

## MYB47 and MYB95 enhance transcriptional activation of *TSA1*

Because *TSA1* gene was not JA-inducible in the *myb47,95* double mutant, we examined the role of each MYB protein in the activation of the *TSA1* promoter using two approaches. First, we examined the binding of MYB47 and MYB95 to the *TSA1* promoter using the electrophoretic mobility shift assay (EMSA). Glutathione *S*-transferase (GST) fusions of MYB47 and MYB95 were separately expressed in *Escherichia coli* (Supplementary Fig. 4), and the purified recombinant proteins were subjected to EMSA. We have identified two conserved sequences in the *TSA1* promoter regions of five Brassicaceae plant species and named these as potential *cis*-elements, box1 (CACGTTT[A/G]) and box2 (GTTAGTT)[20]. The Arabidopsis *TSA1* promoter region, including box1 and box2, was used as the biotinylated DNA probe (hereafter referred to as pTSA1) (Fig. 3a). Both MYB47 and MYB95 bound to the DNA probe in a box2-dependent manner; the addition of unlabeled pTSA1 or unlabeled pTSA1 containing a mutated box1 (pTSA1-m1), but not mutated box2 (pTSA1-m2), reduced the signal of probe–protein complexes (Fig. 3b, c).

Next, a promoter-reporter assay was conducted to examine the effect of MYB47 and MYB95 on the activity of the *TSA1* promoter. The β-glucuronidase (GUS) activity was induced in *Nicotiana benthamiana* leaves when co-inoculated with *Agrobacterium* harboring the *Prom35S:GFP-MYB47* or *Prom35S:GFP-MYB95* construct and *Agrobacterium* harboring the *PromTSA1:GUS* construct (Fig. 3d, Supplementary Fig. 5). Consistent with the in vitro EMSA, a mutation in box2 (*m2-PromTSA1:GUS*) drastically reduced the *TSA1* promoter activity in promoter-reporter assays, despite the presence of MYB47 or MYB95 (Fig. 3d, Supplementary Fig. 5). Unexpectedly, a mutation in box1 (*m1-PromTSA1:GUS*) also reduced the MYB47- and MYB95-dependent activation of the *TSA1* promoter (Fig. 3d,

Supplementary Fig. 5). This suggests that binding of another transcription factor, possibly a bHLH protein, to box1 is required for the *TSA1* promoter, since box1 is proposed as a bHLH binding site[20]. Moreover, mutations in both box1 and box2 elements (*m1/m2-PromTSA1:GUS*) completely abolished the *TSA1* promoter activity (Fig. 3d, Supplementary Fig. 5). These findings indicate that MYB47 and MYB95 activate the *TSA1* promoter in box1- and box2-dependent manner in vivo.

## MYC2, MYC3, and MYC4 regulate the JA-induced expression of *MYB47* and *MYB95*

MYC2, MYC3, and MYC4 transcription factors are central regulators of the JA signaling pathway[37,38], and they are required for the activation of the *TSA1* gene[20]. To examine whether these transcription factors regulate *MYB47* and *MYB95* gene expression, we examined the expression of *MYB47* and *MYB95* genes in the leaves of *myc2 myc3 myc4* (*myc2,3,4*) triple mutant by qRT-PCR. The results revealed that in wild-type leaves, JA treatment strongly and weakly upregulates *MYB95* and *MYB47* expression, respectively, whereas the JA-inducibility of these genes was not observed in *myc2,3,4* mutant leaves (Fig. 4a). By contrast, the expression of *MYC2/3/4* genes was not reduced in the JA-treated leaves of the *myb47,95* double mutant compared to the wild-type (Fig. 4b). These findings indicate that MYC2/3/4 are upstream transcription factors that regulate *MYB47/95* expression in response to JA.

## *MYB47* and *MYB95* Genes Perform Unique Functions in the JA Response

Unlike *TSA1* and *BGLU18*, a JA-responsive gene, *VSP2* was activated by JA at the same levels in the *myb95* single and *myb47,95* double mutants compared to the wild-type (Supplementary Fig. 6). Moreover, JA and coronatine inhibited root elongation in the *myb47,95* double mutant to the same extent as in the wild-type (Supplementary Fig. 6), indicating that the JA signaling pathways required for these responses remain in the *myb47,95* double mutant. This is in contrast with the *myc2,3,4* mutant, where the major JA signaling pathways are blocked (Supplementary Fig. 6)[37].

# a

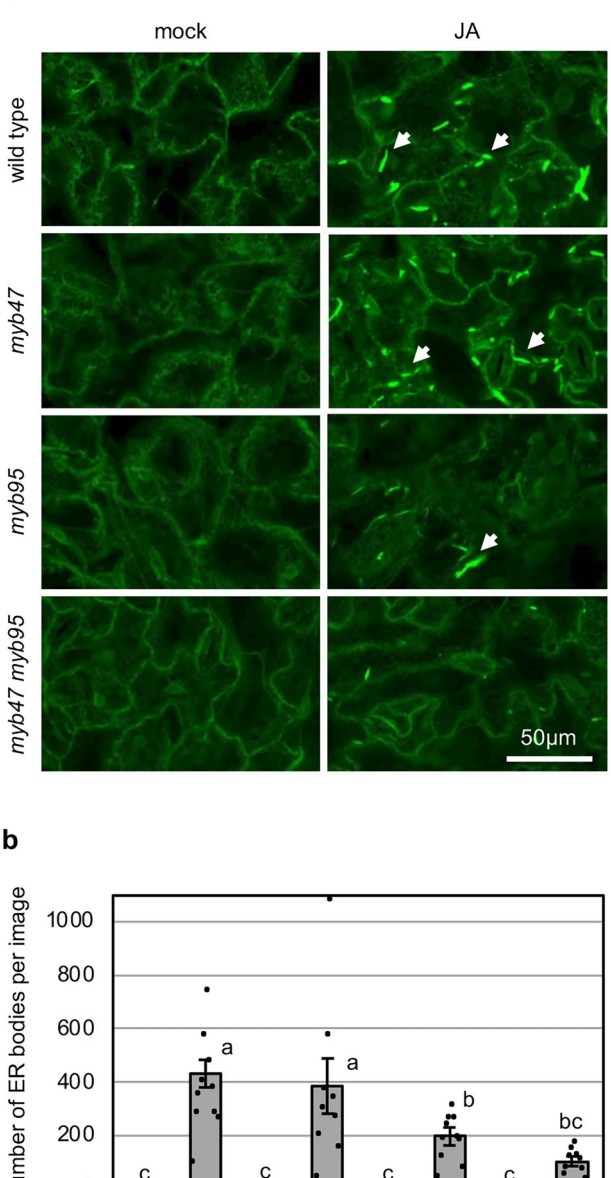

# b

**Fig. 2 | MYB47 and MYB95 are required for the formation of JA-inducible ER bodies. a** Confocal laser scanning microscope images of leaves from stable transgenic plants in which ER and ER bodies are visualized with GFP. Rosette leaves from 14-day-old aseptically grown wild-type (WT), *myb47* and *myb95* single mutants, and *myb47,95* double mutant were cut and floated on distilled water (mock) or 50 μM methyl jasmonate (JA) for 2 days. The GFP signal shows the ER and ER bodies. Arrows represent ER bodies. **b** Number of ER bodies per image (0.32 mm² area). Error bars represent standard error (SE) of 10 images from different leaves (*n* = 10 replicates). Different lowercase letters indicate significant differences (*p* < 0.05; Tukey's test).

These findings support the hypothesis that MYB47 and MYB95 regulate a distinct branch of the JA signaling pathway.

To gain a broader understanding of the roles of *MYB47* and *MYB95* in leaves, we conducted RNA-seq analysis on wild-type, *myc2,3,4*, and *myb47,95* mutant leaves treated with or without JA for two days. In the multidimensional scaling (MDS) plot, the distribution of JA-treated *myc2,3,4* mutant is close to the mock-treated plants, suggesting a reduction of the JA response in *myc2,3,4* mutant (Fig. 5a). By contrast, the

distributions of *myb47,95* double mutant and wild-type showed that these plants have substantial JA responses. However, the distributions of JA-treated *myb47,95* and wild-type differ from each other in the MDS plot, suggesting that the JA response in the *myb47,95* double mutant is distinct from that in wild-type (Fig. 5a). Therefore, we focused on JA-responsive genes upregulated in the wild-type after the JA treatment. A total of 944 genes were JA-responsive [WT (JA) > WT (mock), adjusted *p*-value < 0.05] in wild-type plants (Fig. 5b). Of these, 515 genes showed a reduced JA-response in the *myc2,3,4* mutant [WT (JA) > *myc2,3,4* (JA), adjusted *p*-value < 0.05], indicating that these genes are regulated by MYC2/3/4 transcription factors. Similarly, 77 genes showed a reduced JA-response in the *myb47,95* double mutant [WT (JA) > *myb47,95* (JA), adjusted *p*-value < 0.05], indicating the regulation of these genes by MYB47/95 transcription factors (Fig. 5b). Among the 77 genes that reduced JA response in the *myb47,95* double mutant, 66 showed reduced JA-response in the *myc2,3,4* mutant (Fig. 5b, Supplementary Table 5), suggesting that these genes are directly or indirectly controlled by MYC2/3/4 as well as by the MYB47/95. These 66 genes included *TSA1*, *BGLU18*, and *MEB2* (Supplementary Table 5), which encode protein components of ER bodies.

We integrated our data to examine the gene expression correlation among *MYB47*, *MYB95*, and the 77 reduced-JA-responsive genes in the *myb47,95* double mutant. The results of gene expression correlation analysis revealed several clusters (Supplementary Fig. 7), indicating that genes within a cluster are similarly regulated in response to JA-treated leaves and other experimental conditions. Additionally, these gene clusters included many defense-related genes, suggesting that these cluster genes are involved in JA-mediated plant defense (Supplementary Table 5 and Supplementary Fig. 7). A cluster of *MYB47* and *MYB95* contained *TSA1* and *BGLU18* genes (Fig. 5c), suggesting a strong functional relationship within the cluster. Notably, this cluster included *JAL5*, *JAL6*, *JAL35*, and *At3g28220* (Fig. 5c). *At3g28220* encodes a meprin-associated TRAF homology (MATH) domain-containing protein. JAL and MATH proteins form a large complex with an ER body BGLU23 protein in Arabidopsis seedling homogenate[12,14]. The complex formation of JAL with BGLU23 modifies its enzyme activity in the seedling homogenate[13]. This suggests the possibility that *JAL5*, *JAL6*, *JAL35*, and *At3g28220* genes are involved in the modification of BGLU18 enzyme activity. In contrast to JA-inducible genes, the number of JA-repressed genes controlled by MYB47/95 was small (Supplementary Table 6 and Supplementary Fig. 8).

## Accumulation of seedling ER bodies is not affected in the *myb47 myb95* double mutant

The epidermis of Arabidopsis seedlings accumulates ER bodies, which are developmentally controlled but independent from the JA signaling pathway[9,16,43]. A bHLH transcription factor NAI1 specifically regulates the expression of two genes, *NAI2* and *BGLU23*, whose gene products consequently form the ER bodies in seedlings[5,9,43]. Therefore, we assumed that MYB47 and MYB95 do not contribute to the accumulation of NAI2 and BGLU23 proteins and ER bodies in seedlings. Indeed, we observed that *myb47,95* double mutant accumulates NAI2 and BGLU23 proteins and ER bodies in seedlings at the same levels as the wild-type (Fig. 6). These results indicate that MYB47 and MYB95 merely contribute to the ER body formation in the seedling epidermis.

## Glucosinolate accumulation is not reduced in the *myb47 myb95* double mutant

Because glucosinolates are potential substrates of the ER-body-accumulated BGLUs[5,6], we questioned if MYB47 and MYB95 also contribute to the transcriptional activation of the glucosinolate biosynthesis pathway. We extracted expression data of glucosinolate biosynthetic genes[44] from the transcriptome analysis. The result revealed that most of the glucosinolate biosynthesis genes responded to the JA treatment in the *myb47,95* double mutant, whereas their expression was not JA-inducible in the *myc2,3,4* mutant (Supplementary Fig. 9). These results indicate that MYB47 and MYB95 are not essential for the activation of glucosinolate biosynthesis

**Fig. 3 | MYB47 and MYB95 activate the *TSA1* promoter. a** Arabidopsis *TSA1* promoter harbors conserved sequences (box1 and box2), in addition to the initiator sequence (Inr). The sequences of box1 and box2 in the competitor oligonucleotides (pTSA1, pTSA1-m1, pTSA1-m2, and pTSA1-m1/m2) are shown. Lowercase letters denote mutated nucleotides. **b** Binding of MYB47 to the *TSA1* promoter in vitro. Partially purified GST-MYB47 fusion protein (Supplementary Fig. S3) and a biotinylated-pTSA1 probe were used. **c** Binding of MYB95 to the *TSA1* promoter in vitro. Partially purified GST-MYB95 fusion protein (Supplementary Fig. S3) and a biotinylated-pTSA1 probe were used. **d** Activation of the *TSA1* promoter by MYB47 and MYB95 *in planta*. The chart shows β-glucuronidase/luciferase (GUS/LUC) activity ratios measured 3-day-after the agroinfiltration of *N. benthamiana* leaves with the effector, reporter and *Prom35S:LUC* constructs. Error bars represent SE (*n* = 6 independent experiments). Different lowercase letters indicate significant differences (*p* < 0.05; Tukey's test).

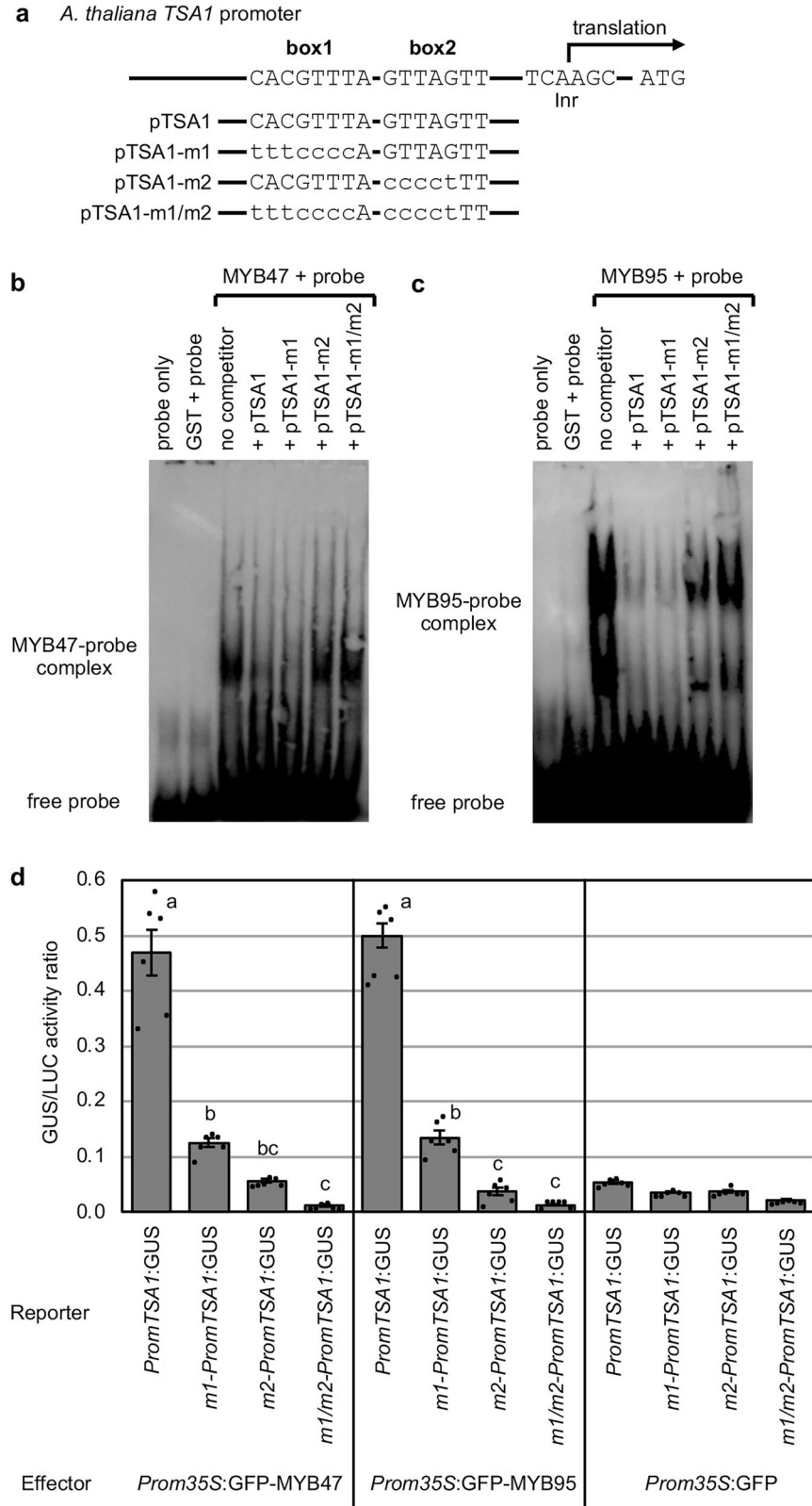

genes. The expression of certain genes was higher in the *myb47,95* double mutant compared to the wild-type, potentially compensating for the loss of the defense mechanisms regulated by MYB47 and MYB95.

To further evaluate the hypothesis that MYB47 and MYB95 are not essential for glucosinolate biosynthesis, we examined the glucosinolate levels in the JA-treated and untreated leaves of the *myb47,95* double mutant, *myc2,3,4* (which is known to be glucosinolate deficient)[39], and the wild-type. Wild-type plants constitutively accumulated aliphatic and indolic glucosinolates regardless of the JA treatment, whereas *myc2,3,4* plants had strongly reduced amounts of aliphatic and indolic glucosinolates. In contrast to the *myc2,3,4* plants, the *myb47,95* double mutant showed no reduction in aliphatic and indole glucosinolate accumulation in leaves (Fig. 7 and

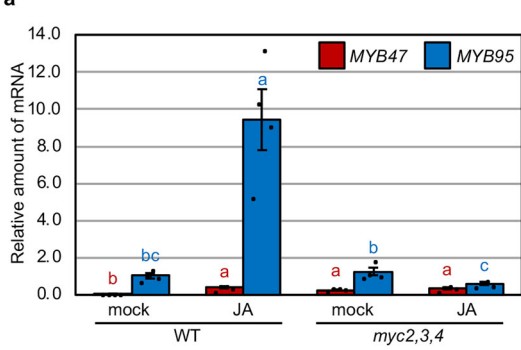

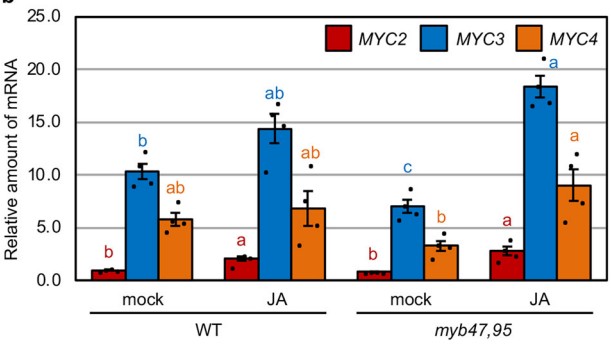

**Fig. 4 | JA response of *MYB47* and *MYB95* genes is reduced in the *myc2,3,4* triple mutant, compared with the normal expression of *MYC2*, *MYC3*, and *MYC4* genes in the *myb47,95* double mutant. a** Expression analysis of *MYB47* and *MYB95* in wild-type (WT) and *myc2,3,4* leaves treated with or without JA by qRT-PCR. Rosette leaves from 14-day-old aseptically grown plants were cut and floated on distilled water (mock) or 50 μM methyl jasmonate (JA) for 24 h. Error bars indicate SE ($n = 4$ independent experiments). Different lowercase letters indicate significant differences ($p < 0.05$; Tukey's test). **b** Expression analysis of *MYC2*, *MYC3*, and *MYC4* in WT and *myb47,95* leaves treated with or without JA for 24 h by qRT-PCR. Error bars indicate SE ($n = 4$ independent experiments). Different lowercase letters indicate significant differences ($p < 0.05$; Tukey's test).

Supplementary Table 7), indicating that MYB47 and MYB95 are not essential for glucosinolate biosynthesis.

Because BGLU18 is a close homologue of BGLU23 that has a myrosinase activity[5,6], we examined whether the glucosinolate breakdown differs between wild-type and *bglu18* or *myb47,95* mutants. We prepared JA-treated rosette leaves to induce BGLU18 accumulation, then homogenized them to promote the enzyme reaction using endogenous glucosinolate substrates. We found that the breakdown of indole-3-ylmethyl glucosinolate (I3M) and 4-methoxyindol-3-ylmethyl glucosinolate (4MOI3M) was reduced in both *bglu18* and *myb47,95* mutants (Supplementary Fig. 10), despite the presence of other major myrosinases known as TGG1 and TGG2[45,46]. These findings suggest that BGLU18 possesses a myrosinase activity.

MYB28/29 and MYB34/51/122 transcription factors mediate aliphatic and indolic glucosinolate biosynthesis, respectively, and the loss of these transcription factors prevents the accumulation of corresponding glucosinolates[28–30,32,33,47]. To investigate whether *TSA1* and *BGLU18* genes are controlled by these MYB transcription factors, we examined *TSA1* and *BGLU18* mRNA levels and their protein levels in the JA-treated leaves of *myb28,29* and *myb34,51,122* plants. The *TSA1* and *BGLU18* gene induction was not reduced in these mutants after JA treatment, and TSA1 and BGLU18 protein levels were similar in these mutants compared to the wild-type (Supplementary Fig. 11), indicating that these MYB transcription factors are not required for the JA-induction of *TSA1* and *BGLU18* genes.

### MYB47 and MYB95 interact with MYC2, MYC3 and MYC4

To determine the nuclear localization of MYB47 and MYB95 proteins, we transiently expressed the *GFP-MYB47* and *GFP-MYB95* fusion proteins in the epidermal cells of *N. benthamiana* leaves (Supplementary Fig. 12). Confocal microscopy analysis revealed a strong GFP signal in the nucleus, in addition to a weak signal in the cytosol (Supplementary Fig. 12), indicating that MYB47 and MYB95 localize to the nucleus.

It has been reported that MYB and MYC transcription factors frequently interact to regulate biological functions[27,48,49]. Prompted by the results of Millard et al.[48], we tested the interactions of MYC2, MYC3 and MYC4 with MYB47 and MYB95 through the yeast two-hybrid assay, in vitro pull-down assay, and *in planta* co-immunoprecipitation assays. The results showed that MYC2, MYC3 and MYC4 interact with MYB47 and MYB95 in all examined systems (Fig. 8, Supplementary Figs. 13, 14). MYC2-MYB95 interaction was weaker than other

**Fig. 5 | Transcriptome analysis reveals that MYB47 and MYB95 regulate a subset of ER-body related genes. a** Overview of the RNA-seq analysis of Arabidopsis leaves. Rosette leaves from 14-day-old aseptically grown wild-type (WT), *myc2,3,4*, and *myb47,95* mutants were cut and floated on distilled water (mock) or 50 μM methyl jasmonate (JA) for 2 days. The graph shows a non-metric multi-dimensional scaling (MDS) plot calculated from the transcriptome data of wild-type (WT; gray), *myc2,3,4* (blue), and *myb47,95* (red) plants treated with (filled circles) or without (empty circles) JA treatment. **b** Venn diagram of JA-response genes whose expressions were controlled by MYC2/3/4 and MYB47/95. **c** Analysis of co-expressed gene cluster. The indicated genes are correlated to each other in the transcriptome co-expression database (ATTED-II). Red circles represent *MYB47*, *MYB95*, *BGLU18*, and *TSA1*. Yellow circles represent genes whose products potentially form a protein complex with BGLU18. Nodes indicate the correlation between co-expressed genes.

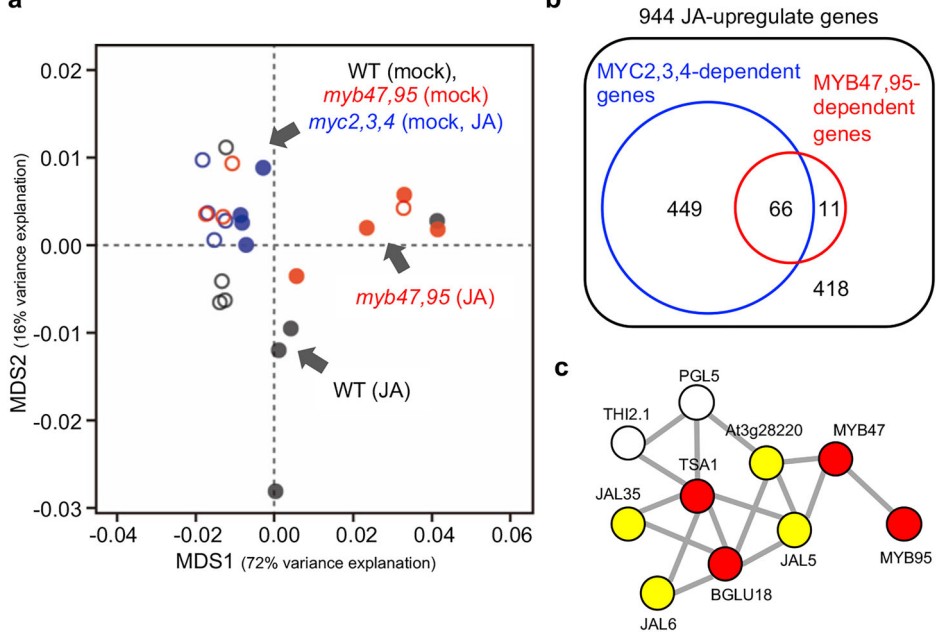

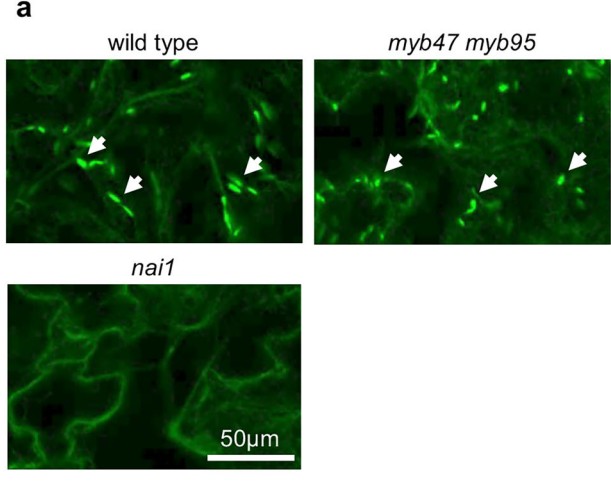

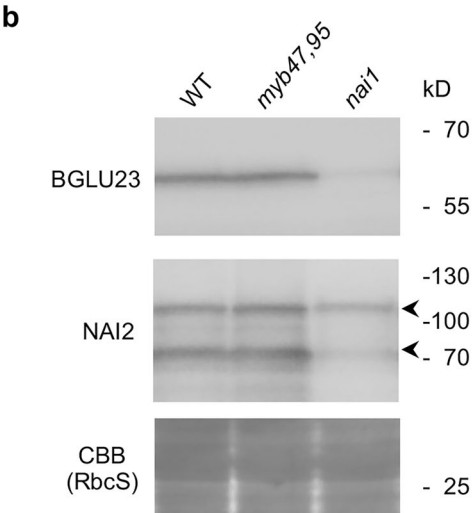

**Fig. 6 | MYB47 and MYB95 are not required for the formation of seedling ER bodies. a** Confocal laser scanning microscope images of cotyledons from 10-day-old stable transgenic plants in which ER and ER bodies are visualized with GFP. The GFP signal shows the ER and ER bodies. **b** Accumulation of BGLU23 and NAI2 proteins in 10-day-old Arabidopsis seedlings. The extracted proteins from seedlings were subjected to immunoblot analysis with anti-BGLU23 antibody or anti-NAI2 antibody. The bands correspond to full length (upper) and partially degraded (lower) NAI2 proteins are shown with arrowheads. Coomassie brilliant blue (CBB) staining shows the RuBisCo large subunit (RbcL) used as a loading control.

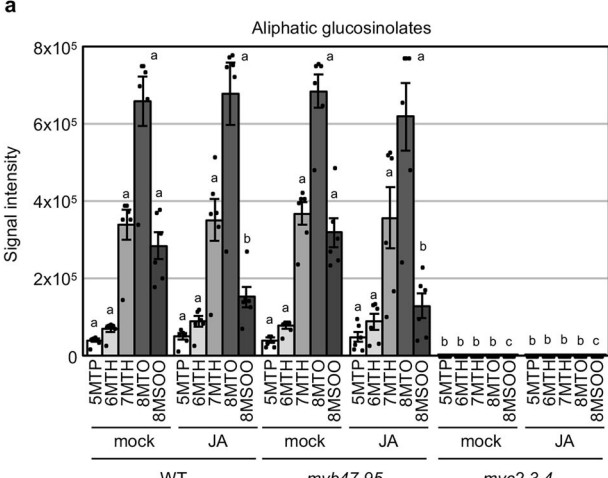

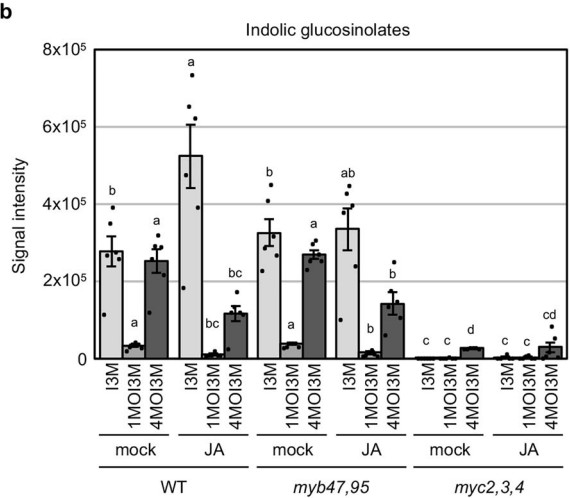

**Fig. 7 | Glucosinolate biosynthesis does not require MYB47 and MYB95.**
**a, b** Levels of aliphatic (**a**) and indolic (**b**) glucosinolates in wild-type (WT), *myc2,3,4*, and *myb47,95* leaves treated with or without JA. Rosette leaves from 14-day-old aseptically grown wild-type (WT), myc2,3,4, and myb47,95 mutants were cut and floated on distilled water (mock) or 50 µM methyl jasmonate (JA) for 2 days. 5MTP, 5-methylthiopentyl; 6MTH, 6-methylthiohexyl; 7MTH, 7-methylthioheptyl; 8MTO, 8-methylthiooctyl; 8MSOO, 8-methylslufinyloctyl; I3M, Indol-3-ylmethyl; 1MOI3M, 1-methoxyindol-3-ylmethyl; 4MOI3M, 4-methoxyindol-3-ylmethyl. Error bars indicate SE (*n* = 6 independent experiments). Different lowercase letters indicate significant differences within the glucosinolate type (*p* < 0.05; Tukey's test).

interactions, whereas MYC3-MYB95 interaction was stronger as judged by β-galactosidase activity in the yeast two-hybrid assay (Fig. 8a). An in vitro pull-down assay confirmed that recombinant MYC2, MYC3, and MYC4 proteins physically interact with GST-fused MYB47 or MYB95 (Supplementary Fig. 14). *In planta* co-immunoprecipitation assays using *N. benthamiana* leaves showed that TagRFP-MYC3 and TagRFP-MYC4 fusion proteins were co-immunoprecipitated with the GFP-MYB47 or GFP-MYB95 fusion protein using anti-GFP antibody (Fig. 8b). The co-immunoprecipitation of MYC2 could not be tested because of its low expression in *N. benthamiana* leaves. Furthermore, GFP-MYB47 and GFP-MYB95 fusions co-localized with TagRFP-MYC3 and TagRFP-MYC4 in condensed structures in the cell when transiently co-expressed in *N. benthamiana* leaves (Supplementary Fig. 15). These data indicate that MYB47 and MYB95 form protein complexes with MYC2, MYC3, and MYC4.

## MYB47 and MYB95 Form Unique Clade in the Phylogenetic Analysis

MYB47 and MYB95 have been reported to be Arabidopsis-specific MYBs in the phylogenetic analysis[26]. To gain an insight into the evolutionary pathway

of these MYBs, we conducted a phylogenetic analysis of MYB47 and MYB95. We selected 53 proteins (9 from gymnosperms, 34 from non-Brassicaceae plants, and 10 from Arabidopsis), which belong to closely related MYB47/95, MYB28/29/76, MYB34/51/122, and MYB16/106 sub-clades, and generated a phylogenetic tree together with unrelated GLAB-ROUS1 and MYB13 (Supplementary Fig. 16). A phylogenetic tree of these proteins showed a separation between the MYB16/106 clade and the other clade containing MYB47/95, MYB28/29/76, MYB34/51/122 (Supplementary Fig. 16). MYB16/106 homologs are involved in epidermal cell shape and wax formation in land plants[50–53], which may explain the wide distribution of MYB16/106 homologs in gymnosperms and many angiosperms. By contrast, MYB47/95, MYB28/29/76, and MYB34/51/122 subclades seem to have evolved exclusively in the Brassicales. Therefore, we reanalysed these protein homologs in Brassicales (Supplementary Fig. 17). A phylogenetic tree of these MYBs shows that none of the three clades is family-specific (Supplementary Fig. 17). These findings suggest that the divergence of the

**Fig. 8 | MYB47 and MYB95 physically interact with MYC2, MYC3, and MYC4. a** Interaction between MYB and MYC proteins in the yeast two-hybrid assay. Constructs expressing the DNA-binding domain (BD)-MYB and activation domain (AD)-MYC fusion proteins were co-expressed in yeast, and protein–protein interactions were examined by measuring β-galactosidase activity. Empty vectors were used as a negative control. Error bars indicate SE ($n = 3$ independent experiments). Different lowercase letters indicate significant differences ($p < 0.05$; Tukey's test). AU, arbitrary unit. **b** Interaction between MYB and MYC proteins *in planta*. Constructs expressing the GFP-MYB and RFP-MYC fusion proteins were transiently expressed in *N. benthamiana* leaves, and input and immunoprecipitated protein complexes were detected by immunoblot analysis. Asterisks indicate nonspecific cross-reactive bands detected using anti-GFP and anti-RFP antibodies.

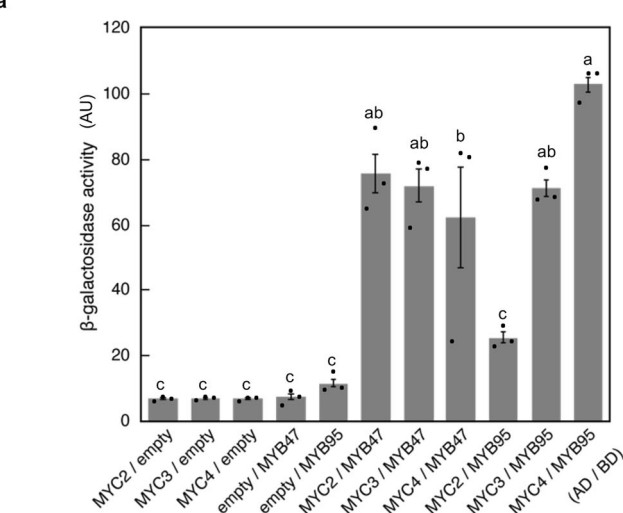

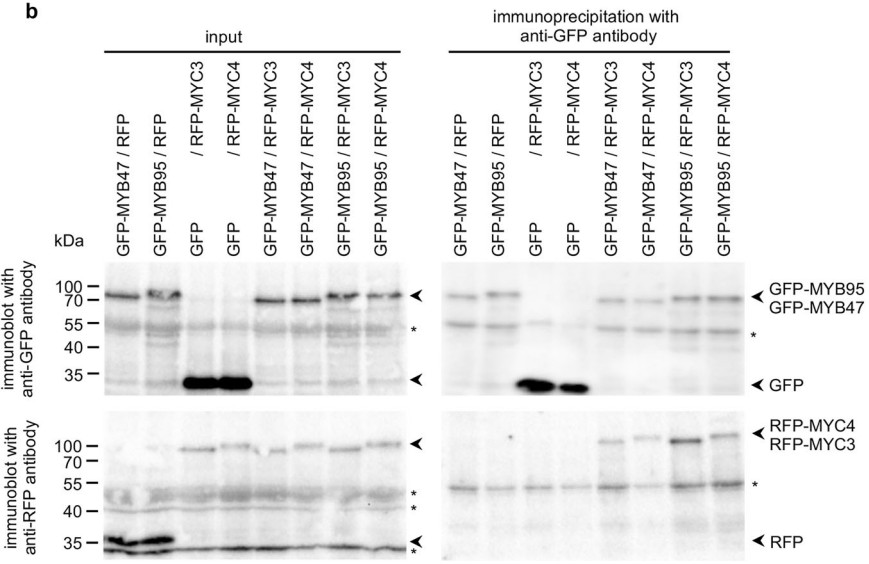

MYB47/95, MYB28/29/76, and MYB34/51/122 subfamilies occurred early during the evolution of the Brassicales plants, in contrast to the ER body formation, which diversified exclusively in the Brassicaceae family[20].

## Discussion

JA treatment induces ER body formation in Arabidopsis rosette leaves, but the regulatory mechanism underlying this phenomenon remains uncharacterized. Here, we identify unique MYB transcription factors that regulate inducible ER-body formation.

Previous works revealed that two homologous ER body scaffold proteins, NAI2 and TSA1 are important for the ER body formation, together with major ER body components such as BGLU23 and BGLU18[5,10,20,21,54]. In Arabidopsis, wounding or JA treatment induces the expression of *TSA1* and *BGLU18*, hence these genes are regarded as the main players of the inducible ER body formation[19–21]. We found that *TSA1* and *BGLU18* expression levels, as well as ER-body accumulation, were reduced in JA-treated *myb95* mutant leaves. Further reduction in *TSA1* and *BGLU18* expression was observed in the *myb47,95* double mutant. Both MYB47 and MYB95 exhibited nuclear localization and recognized *cis*-elements in the *TSA1* promoter to activate its expression. These findings indicate that MYB95 functions as a transcriptional activator of *TSA1* and *BGLU18* genes, and MYB47 supports the role of MYB95 in Arabidopsis leaves. The 5 to 10-day-old seedling epidermis

accumulates ER bodies containing BGLU23 and NAI2, which are regulated by the NAI1 transcription factor[9,10,43]. We found that the *myb47,95* double mutant accumulates BGLU23 and NAI2 proteins, as well as ER bodies, at the same level as the wild-type, indicating that MYB47 and MYB95 do not interfere with developmentally controlled ER body formation in the seedling epidermis. The recent findings indicate that a master regulatory transcription factor of epidermal cell differentiation, namely ATML1, regulates the expression of *NAI1*[17]. In this study, we found that MYC2/3/4, a master regulatory transcription factor of the JA-signaling pathway, controls the expression of *MYB47* and *MYB95*. Therefore, we speculate that there are two regulatory mechanisms for the ER body formation; one is the ATML1-NAI1-mediated pathway that works in the developmentally controlled ER body formation, and the other is the MYC2/3/4-MYB47/95-mediated pathway that promotes ER body formation in response to environmental stress (Fig. 9a).

Arabidopsis *MYB47* and *MYB95* are homologous genes located within a duplicated region on chromosome 1[42], suggesting that these MYB homologs were generated by whole-genome duplication. Amino acid sequences of MYB47 and MYB95 showed high (81%) similarity via BLAST search, suggesting that these proteins perform similar functions. Consistent with this result, both proteins showed similar DNA-binding activity, and both MYB47 and MYB95 proteins similarly activated the *TSA1* promoter

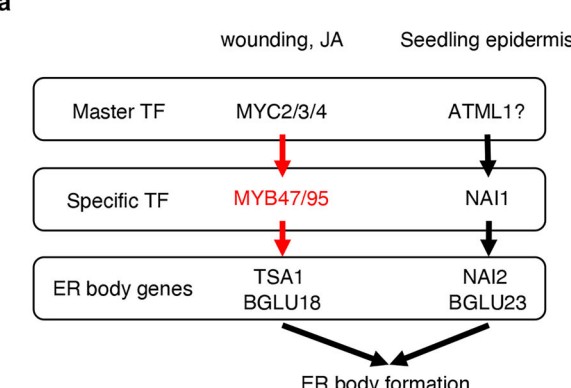

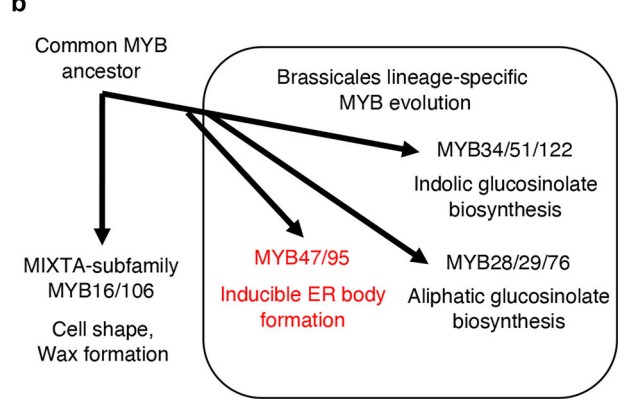

**Fig. 9 | Models of MYB47/95 function. a** A model of functional differentiation in two types of ER body formation. Wounding and JA treatment induces wound- and JA-inducible ER bodies, whose components are TSA1 and BGLU18, whereas the seedling epidermis constitutively accumulates ER bodies, whose components are NAI2 and PYK10/BGLU23. In each ER body formation pathway, there are master transcription factors (MYC2/3/4 and ATML1) that regulate downstream specific transcription factors (MYB47/95 and NAI1), which further regulate the target ER body genes. **b** An evolutionary model of MYB47/95, MYB34/51/122, and MYB28/29/76 that are unique to Brassicales. A common MYB ancestor evolved into MYB16/106, which plays a general function in land plants, whereas MYB47/95, MYB34/51/122, and MYB28/29/76 evolved exclusively during the establishment of the order Brassicales, which play a unique function in these plant lineages.

when overexpressed in *N. benthamiana* leaves. However, ER-body formation, as well as *TSA1* and *BGLU18* gene expression, were strongly affected in the *myb95* mutant compared with the *myb47* mutant after the JA treatment, indicating that MYB95 predominates JA-inducible ER-body formation in Arabidopsis. The JA-inducibility of *MYB95* was higher than that of *MYB47* in Arabidopsis leaves, suggesting that the different expression levels of these transcription factor genes contribute to the phenotypic differences between *myb47* and *myb95* single mutants.

Comparison of the promoter sequences of *TSA1* homologs in the Brassicaceae family revealed two conserved motifs, box1 (CACGTTT[A/G]) and box2 (GTTAGTT), in the region proximal to the start codon[20]. Therefore, we examined whether MYB47 and MYB95 bind to box1 and box2 to regulate *TSA1* expression in Arabidopsis. We found that both transcription factors bind to box2, but not to box1, in vitro. We found that the sequence of box2 is identical to the potential MYB DNA-binding site, which is found in the promoters of defense-related genes in tobacco and tomato[55,56]. In contrast to the in vitro assays, we found that both box1 and box2 elements are essential for the *TSA1* promoter activation by MYB47 and MYB95 in vivo. This finding indicates that box1 may have a role in the regulation of *TSA1* by MYB47 and MYB95. We speculate that MYB47 and MYB95 undergo post-translational modifications or interact with

additional factor(s) to modulate their DNA-binding specificity towards the activation of the *TSA1* promoter in plant cells. Together, these results show that MYB47 and MYB95 regulate *TSA1* expression.

It is well known that MYB and MYC transcription factors physically interact with each other to regulate a large variety of biological processes, including glucosinolate biosynthesis[27,39,49,57,58]. Indeed, we found an interaction between MYC2/3/4 and MYB47/95 in the yeast two-hybrid assay, which is consistent with the results of previous analyses[48,59]. Furthermore, these protein–protein interactions also occurred in vivo, and they co-localized in the cell, suggesting that the MYC and MYB interactions are involved in the regulation of the target gene expression. Millard and colleagues identified a MYC-interaction motif (MIM), the binding site of eight MYB transcription factors unique to Brassicaceae (MYB47/95, MYB28/29/76, and MYB34/51/122) while investigating their interaction with MYC2/3/4[48,59]. Recently, three-dimensional structure analyses revealed the MIM in MYB29 determines the selective MYC-MYB interaction[60]. Considering that these MYBs are conserved among the Brassicaceae plant species[61], it was suggested that their MYC-MYB interactions are evolutionarily conserved in the different processes of chemical defense system (aliphatic and indolic glucosinolate biosynthesis and ER body formation) in Brassicaceae[32,39].

Transcriptome data analysis revealed reduced induction of ER body-related genes (*MEB2, JAL5, JAL6, JAL35,* and *At3g28220*) and defense-related genes in both the JA-treated *myc2,3,4* mutant and *myb47,95* double mutant. However, most of the JA-inducible genes did not show a significant reduction in expression in the *myb47,95* double mutant. The inhibition of root elongation after coronatine or JA treatment in *myb47,95* plants was similar to that observed in the wild-type but different from that observed in the *myc2,3,4* mutant. Collectively, these results indicate that the induction of many JA response genes does not depend on MYB47 and MYB95; instead, these two MYBs may be responsible for the regulation of a subset of the MYC2/3/4 target genes that control plant defense responses such as ER body formation and function. The *myc2,3,4* mutant does not produce JA-inducible ER bodies[20], and the *myc2,3,4* mutant did not show JA-inducible *MYB47* and *MYB95* expression in this study. This suggests that MYC2/3/4 control the formation of JA-inducible ER bodies by regulating *MYB47* and *MYB95* expression and the subsequent induction of *BGLU18* and *TSA1*.

Due to the high sequence similarity between BGLU18 and BGLU23, we speculated that glucosinolates might be substrates of BGLU18. Indeed, we found that I3M and 4MOI3M degradation is partly dependent on BGLU18 in the homogenate of JA-treated leaves. Because ER bodies uniquely appeared in proximity to the wounded sites, it can be concluded that wound-induced ER bodies are involved in defense against microorganism infection or herbivore attack at the wounded sites. Besides this, it has been reported that BGLU18 hydrolyses ABA-glucoside to activate ABA signaling in petioles in non-wounded leaves[62,63]. We did not observe any ABA-related phenotype in our experiments, but it seems that BGLU18 has a dual function in Arabidopsis; one in the activation of ABA-signaling and another in chemical defense.

JA-induced ER body accumulation is inhibited when leaves are treated with ethylene, indicating the antagonistic effect of ethylene on ER-body formation[18]. This phenomenon may involve the downregulation of *MYB95* expression, since the negative regulation of *MYB95* expression by ethylene has been suggested in Arabidopsis[64]. Moreover, the dehydration stress rapidly induces *MYB95* expression[64], whereas green peach aphid (*Myzus persicae*) infection reduces the expression of both *MYB47* and *MYB95* genes[65]. Therefore, the ER body formation is likely to be changed by these stress treatments.

A recent report on the diversification analysis of MYB genes in plants suggests that MYB47 and MYB95 have evolved as a Brassicaceae-specific subclade together with the MYB28/29/76 and MYB34/51/122 subclades[26]. Our phylogenetic analysis also suggests that MYB47/95, MYB28/29/76, and MYB34/51/122 subclades diverged and exclusively evolved during the establishment of Brassicales. These MYBs are responsible for glucosinolate biosynthesis[30,32,39,47] and ER body formation[1,5,7,8], indicating that they play similar functions regarding the unique chemical defense system in

Brassicaceae. Here, we postulate a unique evolutionary model of Brassicales lineage-specific MYBs, suggesting that MYB47/95, MYB28/29/76, and MYB34/51/122 subclades co-evolved with glucosinolate- and ER body-based defense system (Fig. 9b); MYB47/95, MYB28/29/76, and MYB34/51/122 regulate the ER body formation, aliphatic glucosinolate biosynthesis, and indolic glucosinolate biosynthesis, respectively. These regulatory functions, along with the co-evolution of unique MYB-MYC interactions, make these transcription factors unique to the Brassicaceae plants.

## Methods

### Plant Materials and Growth Conditions

*Arabidopsis thaliana* ecotype Columbia (Col-0) and a transgenic line accumulating GFP in its ER and ER bodies (GFP-h) were used as the wild-type. The GFP-h plant was generated by *Agrobacterium*-mediated transformation of Columbia (Col-0) with a *GFP-HDEL* gene that encodes a signal peptide of pumpkin 2S albumin followed by GFP and an ER-retention signal, HDEL[66,67]. T-DNA insertion mutants, *myb47-2* (SALK_123009C), *myb95-1* (GK-314F05), and *myb47-2 myb95-1* (DUPLO_2725, cross of SALK_123009C and GK-314F05)[68] were obtained from the Nottingham Arabidopsis Stock Centre (NASC). The *myb47-2* mutant, which is another allele of the *myb47* mutant used by Renard et al.[35], was characterized by Cao et al.[36]. Homozygous plants were obtained, and gene knockdown of MYB47 and gene knockout of MYB95 were confirmed by genome PCR and qRT-PCR (Supplementary Fig. 2). The primers used for PCR are listed in the Supplementary Table 8. The *GFP-HDEL* gene was introduced into these mutants by *Agrobacterium*-mediated transformation to visualize ER bodies[10]. Previously reported *myc2 myc3 myc4*[69] (a kind gift from H. Frerigmann), *myb28 myb29*[28] (a kind gift from B. A. Halkier), *myb34 myb51 myb122*[69] (a kind gift from H. Frerigmann), *nai1*[9] (a kind gift from I. Hara-Nishimura, also available from NASC, N69075), and *bglu18* (SALK_075731)[19] were used. For the sake of simplicity, the lines are named *myb47, myb95, myb47,95, myc2,3,4, myb28,29,* and *myb34,51,122*. Seeds of all genotypes were germinated aseptically at 22 °C under continuous light (~100 μE·s$^{-1}$·m$^{-2}$) on plates filled with half-strength Murashige and Skoog (1/2 MS) medium (Fujifilm Wako Chemical, Japan) containing 0.4% (w/v) gellan gum (Fujifilm Wako Chemical), 0.5% (w/v) 2-(*N*-morpholino) ethanesulfonic acid (MES)-KOH (pH 5.7), and 1% (w/v) sucrose. Agroinfiltration experiments were conducted using *Nicotiana benthamiana* plants. *N. benthamiana* seeds were germinated in a greenhouse at 22 °C under 16 h light/8-h dark photoperiod.

### JA and Wounding Treatment

For the JA treatment, the primary and secondary leaves of 2-week-old Arabidopsis plants were harvested and floated on the water containing 50 μM of methyl-JA (MeJA; Sigma-Aldrich, St. Louis, USA). The JA treatment lasted for 40–48 h at 22 °C under continuous light. For the wounding treatment, primary and secondary leaves of 2-week-old Arabidopsis plants were harvested and wounded with a 200 μL pipet tip (four sites per leaf). The leaves were floated on the water for 40–48 h at 22 °C under continuous light.

### qRT-PCR Analysis

Total RNAs were isolated from leaves using TRI Reagent (Molecular Research Center, Cincinnati, USA)[20]. The RNA was dissolved in distilled water, and the primary cDNA strand was synthesized from 1 to 2 μg of the RNA by using Ready-to-Go RT-PCR beads (Cytiva) with random oligomers[20]. Real-time PCR was performed using the QuantStudio 6 (Thermo Fisher Scientific) and a PowerUp SYBR Green Master Mix (Thermo Fisher Scientific) according to the manufacturer's instructions[20]. The primers used for qRT-PCR are listed in the Supplementary Table 8. Target mRNA levels were first normalized to endogenous control genes, and then compared[20]. UBQ10 mRNA levels were used as endogenous control genes[20]. We examined the amplification efficiency of all primer sets. The amplification efficiencies were from $2^{0.8}$ to $2^{1.1}$ per cycle. We estimated

relative mRNA levels of target genes in comparison to the control gene from the PCR-cycle time and amplification efficiency, with the amplification efficiency calibrated calculation method[70].

### Immunoblot Analysis

To perform immunoblot analysis, 100 mg of *N. benthamiana* leaves were homogenized with 200 μL of 2× sodium dodecyl sulfate-polyacrylamide gel electrophoresis (SDS-PAGE) sample buffer containing 1 mM phenylmethylsulfonyl fluoride (PMSF) and 100 μM E-64d (Merck). Then, 5 μL of the extract was used for immunoblot analysis.

The protein samples were separated by SDS-PAGE and then transferred onto a polyvinylidene difluoride (PVDF) membrane for immunoblot analysis. Rabbit anti-NAI2/ΔSP antibody (1:2000 dilution) and rabbit anti-BGLU18 antibody (1:2000 dilution) were used to detect TSA1, NAI2 and BGLU18, respectively[10,19,20]. Mouse anti-GFP (JL-8; Takara Bio, Japan; 1:2000 dilution), mouse anti-tagRFP (RF5R; Agrisera; 1:1500 dilution), mouse anti-His (His.H8; Thermo-Fisher; 1:2000 dilution), and mouse anti-GST (8-326; Thermo-Fisher; 1:1500 dilution) antibodies were used in the immunoblot in the pull-down and co-immunoprecipitation assays. Rabbit anti-PYK10 antibody was generated against insect cell expressed recombinant PYK10 protein and used as 1:2000 dilution. Horseradish peroxidase (HRP)-conjugated mouse anti-IgG (1:4000 dilution) and rabbit anti-IgG (1:2000 dilution) antibodies were used as the secondary antibodies. Signals were detected with enhanced chemical luminescence (SuperSignal West Femto, Thermo-Fisher) using a CCD camera (ChemiDoc, Bio-Rad).

### Microscopy

A confocal laser scanning microscope (LSM880; Carl Zeiss) was used to observe fluorescent proteins. An argon laser (488 nm) and a band-pass filter (493/573 nm) were used to detect the GFP signal, whereas a diode-pumped solid-state laser (561 nm) and a band-pass filter (579/641 nm) were used to detect the RFP signal.

### Quantification of ER Bodies

Arabidopsis leaves were fixed in a solution containing 25 mM sodium phosphate (pH 7.2), 100 mM sucrose, 4% (w/v) paraformaldehyde, and 0.5% (v/v) glutaraldehyde to stop the movement of ER bodies. The z-stack images of the epidermal layer of leaf samples were captured with the confocal laser scanning microscope. The exposure time was reduced to optimize capturing ER bodies but not ER networks. All images were taken with the same settings in the exposure time. The z-stack images were projected into a single-plane image with a maximum intensity method, and ER bodies were counted with the ImageJ software. The noise and background signals were removed by thresholding. ER bodies were extracted by Analyze Particles function and counted. All images were processed with the same settings.

### RNA-seq Analysis

Total RNA was isolated from 50 mg of Arabidopsis leaves using the Mag-MAX Plant RNA Isolation Kit (Thermo-Fisher) and then treated with DNase I (Sigma-Aldrich) to remove traces of contaminating DNA. Construction of the RNA-seq libraries and next-generation sequencing were performed by Macrogen (Seoul, South Korea). Standard libraries were prepared using the TruSeq RNA Library Prep Kit v2 (Illumina). Sequencing was performed on the NovaSeq6000 platform (Illumina) to generate 150 bp paired-end reads. Raw sequence reads (~40 million per sample) were preprocessed using *fastqc* (v0.11.8), with default settings for paired-end reads. The *trimgalore* (v0.6.5) pipeline was used to remove the adaptor sequences, and the clean reads were pseudo-aligned to the *Arabidopsis thaliana* Col-0 transcriptome reference (TAIR10) using the *salmon* (v1.1.0) pipeline. After the removal of low-abundance transcripts not present within replicates under each condition, the count data were imported using the *tximport* (v1.12.3) package. Differential expression analyses were performed using the *edgeR* (v3.26.5) package. The Benjamin-Hochberg method was used to calculate the adjusted *p*-value for multiple comparisons.

## Plasmid Construction

Coding sequences (CDSs) of *MYC2*, *MYC3*, *MYC4*, *MYB47*, and *MYB95* were amplified from Arabidopsis Col-0 cDNA by PCR using sequence-specific primers (Supplementary Table 8), and the PCR products were cloned into the *pENTR/SD/D-TOPO* entry vector (Thermo-Fisher).

To perform the yeast two-hybrid assay, CDSs of *MYC2*, *MYC3*, and *MYC4* were transferred from the respective entry clones into the *pGAD-C1GW3* vector[71] using the LR clonase (Thermo-Fisher) to generate *pGAD-C/MYC2*, *pGAD-C/MYC3*, and *pGAD-C/MYC4* constructs, which encode the activation domain (AD) fusions of MYC proteins. These vectors were made Gateway compatible using the Gateway conversion kit. Additionally, CDSs of *MYB47* and *MYB95* were transferred from the respective entry vectors into *pGBD-C1GW3*[71] with LR clonase to generate *pGBD-C/MYB47* and *pGBD-C/MYB95* constructs, which encode the DNA-binding domain (BD) fusions of MYB proteins.

To perform the in vitro pull-down assay, CDSs of *MYB47* and *MYB95* were transferred from the respective entry vectors into the *pDEST15* vector (Thermo-Fisher) with LR clonase, generating *pDEST15/MYB47* and *pDEST15/MYB95* constructs, which encode GST-MYB fusion proteins. Then, CDSs of *MYC2*, *MYC3*, and *MYC4* were transferred from the respective entry vectors into *pET32a-GW*[72] with LR clonase to generate *pET32a/MYC2*, *pET32a/MYC3*, and *pET32a/MYC4* constructs, which encode TRX-6×His-MYC fusion proteins.

The Gateway binary vector *pGWB406* (a gift from T. Nakagawa) was used for constructing *Prom35S:GFP-MYB47* and *Prom35S:GFP-MYB95*. The CDSs of *MYB47* and *MYB95* were transferred from the respective entry vectors into the *pGWB406* destination vector with LR clonase, thus generating the *pGWB406/MYB47* and *pGWB406/MYB95* vectors, respectively. Similarly, the Gateway binary vector *pGWB561* (a gift from T. Nakagawa) was used for constructing *Prom35S:tagRFP-MYC2*, *Prom35S:tagRFP-MYC3*, and *Prom35S:tagRFP-MYC4*. The CDSs of *MYC2*, *MYC3*, and *MYC4* were transferred from the respective entry vectors into the pGWB561 destination vector to generate *pGWB561/MYC2*, *pGWB561/MYC3*, and *pGWB561/MYC4* vectors, respectively.

The *TSA1* promoter sequence (~635 bp) was amplified from the genomic DNA of Arabidopsis ecotype Col-0 by PCR using pTSA1HindIIIF and pTSA1BamHIR primers (Supplementary Table 8). The PCR product was cloned into the pBI121 binary vector using the *Hin*dIII and *Bam*HI restriction endonucleases to generate the *pBI121/PromTSA1:GUS* construct. Two mutated versions of the *TSA1* promoter were generated by overlap extension PCR using pTSA1HindIIIF/mbox2-pTSA1-R and pTSA1BamHIR/mbox2-pTSA1-F primer pairs (Supplementary Table 8). PCR products of the two reactions were pooled and subjected to another round of PCR using the pTSA1HindIIIF/pTSA1BamHIR primer pair. The mutated versions of the *TSA1* promoter were cloned into the pBI121 vector using the *Hin*dIII and *Bam*HI enzymes to generate *pBI121/m1-PromTSA1:GUS*, *pBI121/m2-PromTSA1:GUS*, and *pBI121/m1/m2-PromTSA1:GUS* plasmids.

## Yeast Two-hybrid Assay

The AD and BD plasmids were introduced into the yeast (*Saccharomyces cerevisiae*) strain AH109 (Takara Bio). The transformed yeast cells were grown in liquid synthetic dropout (SD) medium lacking leucine and tryptophan (SD/-Leu/-Trp) at 28 °C for 48 h, and the concentration of the cell suspension was adjusted until its optical density at 600 nm ($OD_{600}$) reached a value of 0.6. Then, 5 μL of each cell suspension was plated on SD medium lacking adenine, His, Leu, and Trp (SD/-Ade/-His/-Leu/-Trp) and supplemented with 50 mM 3-aminotriazole and 20 mg/L 5-bromo-4-chloro-3-indolyl-β-*D*-galactopyranoside (X-Gal). Plates were incubated at 28 °C, and protein–protein interactions were observed after 8 days. Galactosidase activity in the yeast two-hybrid assay was determined using the Yeast β-Galactosidase Assay Kit (Thermo-Fisher) and following the user manual. Yeast cells grown on SD/-Leu/-Trp medium for 2 days were suspended in water, and the cell concentration was measured at $OD_{600}$. The suspension was mixed with an equal volume of assay solution and incubated at room temperature for 2 h. The enzymatic reaction was detected with $OD_{420}$ in the supernatant, and the activity was calculated according to the user manual.

## Recombinant Protein Production

The recombinant GST-MYB47 and GST-MYB95 fusion proteins were produced in *E. coli* Rosetta 2 (DE3) cells (Merck) using the ZYP-5052 auto-induction medium, which contains terrific broth (TB) supplemented with 1 M $MgSO_4$, 50× 5052 (0.5% [w/v] glycerol, 0.05% [w/v] glucose, and 0.2% [w/v] lactose), 20× NPS (0.5 M $(NH_4)_2SO_4$, 1 M $K_2HPO_4$, and 1 M $NaH_2PO_4$)[73]. Cells harboring the *pDEST15/MYB47* or *pDEST15/MYB95* plasmid were precultured in TB medium and then transferred to the induction medium. To produce GST-MYB47, cells were cultured at 30 °C overnight. To produce GST-MYB95, cells were first cultured at 37 °C for 4–5 h (to obtain $OD_{600}$ > 1.0) and then at 30 °C overnight. The cells were then harvested and stored at −80 °C.

The TRX-MYC2 fusion protein was expressed using *E. coli* BL21 (DE3) pLysS Rosetta cells (Merck) in TB medium supplemented with 1% (w/v) glucose, and TRX-MYC3 and TRX-MYC4 were produced using BL21 (DE3) cells in TB medium supplemented with 3% (v/v) ethanol. Cells harboring the *pET32aGW/MYC2*, *pET32aGW/MYC3*, or *pET32aGW/MYC4* plasmid were precultured in TB medium and then transferred to the respective induction medium, which was incubated at 37 °C until reaching an $OD_{600}$ value of 0.5–0.7. The production of recombinant proteins was induced by culturing the cells with 0.5 mM isopropyl β-D-thiogalactoside (IPTG) at 18 °C for 18 h. The cells were then harvested and stored at −80 °C.

## Recombinant GST-fusion Protein Purification

*E. coli* cells expressing the GST-MYB47 and GST-MYB95 fusion proteins were suspended in the extraction buffer (CelLytic B buffer [Merck] containing 0.1% [w/v] lysozyme, 0.01% [w/v] DNase I, and 1 mM PMSF), incubated on ice for 30 min, and lysed by sonication. Subsequently, the lysate was mixed with an equal volume of 1× phosphate buffered-saline (PBS) containing 0.1% (v/v) Triton X-100 at 4 °C for 30 min, and the insoluble bacterial debris was removed by centrifugation. Then, DTT was added to the supernatant at a final concentration of 1 mM, and the soluble fractions were incubated with glutathione-sepharose 4B beads (GE Healthcare) at 4 °C for 1.5 h with gentle rotation. The beads were transferred to a column and subsequently washed three times with 10× volume of PBS containing 0.1% (v/v) Triton X-100. The GST-MYB fusion proteins were eluted with a buffer containing 50 mM Tris-HCl (pH 8.0), 200 mM NaCl, and 20 μM glutathione. The eluant containing GST-MYB47 was subjected to ultrafiltration with Amicon Ultra Centrifugal Filter Unit Ultracel (3 K MWCO, Merck) by exchanging the buffer with the dialysis buffer (1× PBS, 1 mM DTT, 5% [v/v] glycerol). The eluant containing GST-MYB95 was subjected to dialysis with the dialysis buffer at 4 °C, and then concentrated by ultrafiltration.

## Agroinfiltration

*Agrobacterium tumefaciens* strains EHA105 and GV3101(pMP90RK) were precultured in lysogeny broth (LB) medium and then transferred to an induction broth medium (1 g/L $NH_4Cl$, 0.3 g/L $MgSO_4·7H_2O$, 0.15 g/L KCl, 10 mg/L $CaCl_2$, 2.5 mg/L $FeSO_4·7H_2O$, 2 mM sodium phosphate [pH 7.2], 20% [w/v] glucose, 20 mM MES-KOH buffer [pH 5.5], and 100 μM acetosyringone) supplemented with appropriate antibiotics[74]. *Agrobacterium* cells were resuspended in the infiltration medium (0.5% [w/v] glucose, 10 mM MES-KOH [pH 5.5], 10 mM $MgCl_2$, and 200 μM acetosyringone) ($OD_{600}$ = 1.0). *Agrobacterium* cells carrying the *p19* silencing suppressor (a gift from W. Strzałka) were resuspended in the infiltration medium ($OD_{600}$ = 0.5). Equal volumes of these *Agrobacterium* cell suspensions were mixed and infiltrated into the leaves of 4–5-week-old *N. benthamiana* plants[11].

## Pull-down Assay and Co-immunoprecipitation Assays

Glutathione-Sepharose 4B beads bound by GST, GST-MYB47, or GST-MYB95 were prepared and incubated with the bacterial extracts containing

TRX, TRX-MYC2, TRX-MYC3, or TRX-MYC4 to allow protein complex formation. The GST-fusion proteins were extracted from the bacteria by sonication using the lysis buffer (50 mM Tris-HCl [pH 7.5], 300 mM NaCl, 5% [v/v] glycerol, 1 mM dithiothreitol [DTT], 1 mM PMSF, and protease inhibitor cocktail [Thermo-Fisher]). The lysates were centrifuged at 18,000 × *g* for 30 min, and the supernatants were incubated with 50% (v/v) slurry of Glutathione-Sepharose 4B beads pre-equilibrated in the binding buffer (50 mM Tris-HCl [pH 7.5], 300 mM NaCl, 1 mM DTT, 10% [v/v] glycerol, and protease inhibitor cocktail) at 4 °C for 4 h. Then, the beads were washed with a wash buffer (50 mM Tris-HCl [pH 7.5], 300 mM NaCl, 5% [v/v] glycerol and 1 mM DTT). The TRX-fusion proteins were extracted from the bacteria with the lysis buffer. The lysates were centrifuged, and supernatants were gently mixed with the beads at 4 °C overnight. The beads were washed several times with the wash buffer, and binding proteins were released off the beads with 1× SDS-PAGE sample buffer by heating the samples at 100 °C for 10 min. The eluted proteins were separated by SDS-PAGE and then subjected to immunoblot analysis.

To perform the in vivo co-immunoprecipitation assay, constructs expressing the GFP-MYB and tagRFP-MYC fusion proteins were transiently co-expressed in *N. benthamiana* leaves via the agroinfiltration method. Approximately 1 g of agroinfiltrated leaves were homogenized in liquid nitrogen, and proteins were extracted with the extraction buffer (50 mm Tris-HCl [pH 7.5], 150 mm NaCl, 2.5 mM EDTA, 0.2% [v/v] Triton X-100, 0.1% [v/v] Nonidet P-40, 5% [v/v] glycerol, 1 mM DTT, 2% [w/v] polyvinylpolypyrrolidone, 0.05% [w/v] sodium deoxycholate, 100 μM E-64d, and 1 mM PMSF; 2.5 μL per mg fresh weight). The homogenate was incubated on ice for 20 min, and then sonicated using three 5 s pulses at 10% power. Samples were incubated on ice for 10 min. Then, the lysates were subjected to centrifugation twice for 15 min each at 16,000 × *g* and 4 °C. The supernatants were filtered through two layers of miracloth (Merck). A protein complex containing the GFP-MYB fusion proteins was co-immunoprecipitated using the μMACS GFP isolation kit (Miltenyi Biotec, Germany). The purified protein samples were separated by SDS-PAGE and then analyzed by immunoblotting.

## EMSA
A 36 bp biotinylated double-stranded DNA probe and competitor DNAs (Supplementary Table 8) were generated by annealing a 50 μM mixture of two complementary oligonucleotides (Supplementary Table 8) in 10 mM Tris-HCl (pH 8.0), 50 mM NaCl, and 1 mM EDTA by heating to 95 °C for 5 min and then slowly cooling to room temperature[11]. The purified GST, GST-MYB47, or GST-MYB95 protein (500 ng) was incubated with the DNA probe in the binding buffer (10 mM Tris-HCl [pH7.5], 50 mM KCl, 1 mM DTT, 2.5% [v/v] glycerol, 5 mM MgCl₂, 0.1% [w/v] IGEPAL CA-630, and 0.05 μg/μL poly(dI-dC)) on ice. Competitor oligonucleotides (20-fold excess relative to the probe) were added to the binding reaction, and incubated at room temperature for 20 min. Following the addition of the pTSA1 probe at a final concentration of 5 μM, the mixture was further incubated at room temperature for 30 min, and the DNA–protein complexes were separated by PAGE on 5% (w/v) acrylamide/bis-acrylamide gel in 0.5× Tris/borate/EDTA (TBE) buffer. Samples were electrophoretically transferred from the gel on to a nylon membrane (Hybond-N⁺, Merck). The membrane was blocked with 3% (w/v) skim milk, and biotinylated DNA probe was detected with Avidin-HRP conjugate (Thermo-Fisher) by enhanced chemical luminescence (SuperSignal West Femto, Thermo-Fisher).

## Promoter-reporter transactivation assay
*Agrobacterium* cells harboring the effector construct (*Prom35S:GFP*, *Prom35S:GFP-MYB47*, or *Prom35S:GFP-MYB95*), reporter construct (*PromTSA1:GUS*, *m1-PromTSA1:GUS*, *m2-PromTSA1:GUS*, or *m1/m2-PromTSA1:GUS*), and luciferase gene construct (*Prom35S:LUC*) were co-inoculated into the leaves of *N. benthamiana* plants. After 3 days, the inoculated leaves were harvested and homogenized in pre-cooled CCRL buffer (Promega). The samples were centrifuged twice for 15 min at

10,000 rpm at 4 °C. For the measurement of GUS activity, aliquots of each supernatant (25 μL) were incubated with 75 μL of 10 mM 4-methy-lumbelliferyl-β-D-glucuronide, 0.1% (w/v) sodium lauryl sarcosine, and 10 mM β-mercaptoethanol in the CCRL buffer at 37 °C for 0 min (control) or 30 min[75]. The reactions were terminated with 180 μL of 0.2 M Na₂CO₃. The amount of the product (7-hydroxy-4-methylcoumarin) was measured using a microplate reader (Infinite 200, Tecan, Switzerland), at excitation and emission wavelengths of 355 and 460 nm, respectively. Aliquots of the above supernatants were also used to determine LUC activity using a LUC assay system kit (Promega). Protein concentration in each sample was measured with the Bradford method.

For the GUS staining, leaves were cut into small pieces and vacuum-infiltrated in the GUS staining buffer (100 mM sodium phosphate [pH 7.0], 1 mM 5-bromo-4-chloro-3-indolyl-β-D-glucuronide, 0.5 mM potassium ferrocyanide, 0.5 mM potassium ferricyanide, and 10 mM EDTA) for 5 min[75]. The leaf samples were incubated in the GUS staining buffer at 37 °C for 16 h. The stained leaf tissues were then rinsed with ethanol and water to remove chlorophyll.

## Measurement of Glucosinolate Content
The JA-treated or untreated primary and secondary leaves of 2-week-old plants were harvested and frozen in liquid nitrogen. The frozen leaves (~200 mg) were homogenized in dimethyl sulfoxide (DMSO; 2.5 μL per mg fresh weight) and centrifuged. The supernatant (~200 μL) was collected and analyzed by ultra-performance liquid chromatography-tandem mass spectrometry (UPLC-MS/MS) using the Acquity UPLC instrument (Waters, USA) coupled with a micrOTOF-Q mass spectrometer (Bruker Daltonics, Germany)[76]. Chromatographic separations by UPLC were carried out on XSelect HSS T3 column (2.1 × 150 mm, 2.5 μm particle size) (Waters, USA) using 0.1% solution of formic acid in water (solvent A) and 98% acetonitrile in water (solvent B) with mobile phase flow of 0.3 ml/min in the following gradient[76]. The MS spectra were recorded in negative ionization mode and scanned in the range 50–1000 m/z[76]. The obtained raw data were processed using the MZmine v2.32 software[77]. The extracted ion chromatograms were deconvoluted, aligned, and gap-filled. Finally, the data matrix was limited to chromatograms of molecular ions corresponding to glucosinolates[76]. The amounts of specific glucosinolates were estimated by calculating the respective peak areas in the molecular ion chromatograms.

## Bioinformatics Analysis
Gene co-expression analysis was performed using ATTED-II (http://atted.jp), and GWAS data were analyzed using the GWA-Portal (https://gwas.gmi.oeaw.ac.at). The expression of *BGLU18* and *TSA1* in Arabidopsis leaves was analyzed using the data publicly available in the National Center for Biotechnology Information (NCBI) GEO database (https://www.ncbi.nlm.nih.gov/geo/) under the accession number GSE80744[41].

For the phylogenetic tree analysis, we gathered protein sequences of subfamily VIII-D MYB homologs in Putative Orthologous Groups DB (http://pogs.uoregon.edu) by using Arabidopsis MYBs as a query. To gather MYB sequences in wide ranging plant species, we performed a BLAST search in the NCBI/GenBank database (https://blast.ncbi.nlm.nih.gov) with three restricted database settings: Acrogymnosperm, Amborella and non-Brassicaceae. All protein sequences were merged, the redundancy was reduced, and truncated proteins were removed. The extracted protein sequences were subjected to phylogenetic analysis, and the proteins included in the clade of MYB16/106, MYB47/95, MYB28/29/76, and MYB34/51/122 were used for a dataset of Supplementary Fig. 16. Apart from this, we also searched for Brassicales MYB16, MYB47, MYB28, and MYB34 homologs from the 1000 Plant (1KP) transcriptome database (https://db.cngb.org/onekp/) by BLAST searching and used the results for a dataset of Supplementary Fig. 17. The phylogenetic analysis was performed with a method according to Poncet et al.[78]. Multiple sequence alignments were performed using MUSCLE, and well-aligned sites were chosen using GBlocks. Maximum-likelihood reconstructions incorporating 500 bootstrap

replicates were performed based on the JTT matrix-based model in MEGA-X. The phylogenetic trees were drawn with iTOL (https://itol.embl.de).

## Statistics and reproducibility

We introduced statistical analysis (Tukey's multiple comparison test or Mann-Whitney U test) for all countable data with 3 to 10 biological replications. We observed multiple cells and show representatives for confocal microscope images. Experiments using immunoblots were performed two to four times to validate the reproducibility.

## Reporting summary

Further information on research design is available in the Nature Portfolio Reporting Summary linked to this article.

## Data availability

Sequence data from this article can be found in the Arabidopsis Genome Initiative (AGI) or GenBank/EMBL databases under the following accession numbers: MYB47 (At1g18710), MYB95 (At1g74430), BGLU18 (At1g52400), TSA1 (At1g52410), MYC2 (At1g32640), MYC3 (At5g46760), and MYC4 (At4g17880). The accession numbers of the T-DNA insertion mutants are as follows: *myb47-2* (SALK_123009C), *myb95-1* (GK-314F05), *myb47-2 myb95-1* (DUPLO_2725), and *bglu18* (SALK_075731). The RNA-seq data were deposited in the NCBI GEO database under the accession number GSE288027. Accession numbers of proteins used for phylogenetic analysis are listed in Supplementary Table 9. The source data for figures and processed RNA-seq data are available as Supplementary Data 1 and 2, respectively. Uncropped blot and gel images are available in Supplementary Fig. 18. The materials used in this study are available from the corresponding author upon request.

## Code availability

R scripts for the RNA-seq data analysis are available on GitHub (https://github.com/arpankbasak/RNASeq_analysis_MYB47MYB95.git).

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

## Acknowledgements

We thank Shoji Mano (National Institute for Basic Biology) and Tsuyoshi Nakagawa (Shimane University) for providing the vectors; Henning Frerigmann (Max Plank Institute for Plant Breeding Research) and Barbara Ann Halkier (Copenhagen University) for sharing the Arabidopsis mutants; Rościsław Krutyhołowa (Jagiellonian University) for sharing the GST protein; Wojciech Strzałka (Jagiellonian University) for providing an *Agrobacterium tumefaciens* strain; Karolina Małek (Jagiellonian University) for supporting laboratory; MCB core facilities for instrumentation. This work was supported by the National Science Centre of Poland (OPUS grant UMO-2016/23/B/NZ1/01847 to A.K.B. and UMO-2020/37/B/NZ3/04176 to K.Y.), and the Foundation for Polish Science (TEAM grant TEAM/2017-4/41 to S.S.) and by the institutional support provided by the Malopolska Centre of Biotechnology, Jagiellonian University.

## Author contributions

K.Y. conceived the study and supervised the project; J.B., S.S., A.K.B., S.B., S.G.-Y., K.E., M.M., P.C., P.B., and K.Y. designed the experiments; J.B., S.B., K.E., M.M. K.T.-G., and K.Y. performed qRT-PCR and immunoblot; J.B., S.B., and S.G.-Y. performed yeast-two hybrid assay; S.S., and S.B. performed EMSA, promoter-reporter assay, pull-down assay, and co-immunoprecipitation assay; K.Y. performed the microscopic analysis and phylogenetic tree analysis; J.B., A.K.B., S.B., P.C., and P.B. performed the glucosinolate analysis; A.K.B. and K.Y. performed the RNA-seq data interpretation; S.S., K.T.-G., R.B., K.B., P.B., and K.Y. wrote the manuscript; All authors approved the final version of the manuscript.

## Competing interests

The authors declare no competing interests.
