## [Transparent Peer Review file · Communications Biology]

Arabidopsis MYB47 and MYB95 Transcription Factors Regulate Jasmonate-inducible ER-Body Formation

Corresponding Author: Dr Kenji Yamada

This manuscript has been previously submitted at another journal. This document only contains information relating to versions considered at Communications Biology.

Version 0:

Reviewer comments:

Reviewer #1

(Remarks to the Author)

The ms by Bizan et al describes a comprehensive study of MYB47 and MYB95, two homologous transcription factor genes that are specific to Brassicales. The authors use a set of genetic, molecular, microscopic and metabolic tools to establish that MYB47/95 are mediating the expression of a subset of MYC2/3/4-dependent JA-inducible genes, and whose protein products are integral components of ER-bodies, a type of endoplasmic reticulum-derived subcellular structure mediating defensive functions in Brassicaceae against insect pests and pathogenic fungi. They come-up with a model of lineage-specific MYB evolution where the widespread JA mediators MYC234 direct the JA-induced expression of Brassicaceae-evolved MYB transcription factors MYB47/95. MYB47/95 proteins interact directly with MYC234 in vitro and in vivo, presumably to mediate specific target expression. MYB47/95 then positively control the specific expression of 66-77 genes; one of them, TSA1 was used as an example to demonstrate experimentally MYB47/95 binding to its promoter. Many of the target proteins are required for ER-body formation, but not for glucosinolate biosynthesis, which is directed by two other sets of MYB TFs.

The work is original and interesting, it convincingly assigns a new function to two neglected MYB TFs, and depicts a Brassicaceae-specific transcriptional pathway for the JA-induced biogenesis of ER-bodies. The experiments are overall well and logically conceived but the manuscript suffers a number of approximations in its presentations. In some instances some informations are missing and need further experiments.

Major concerns:

1. Although little is known on these genes/proteins, two previous papers have described MYB47 involvement in two different biological contexts: Renard et al (<https://pubmed.ncbi.nlm.nih.gov/32519347/>) have shown MYB47 to be required for seed longevity; Marquis et al (<https://pubmed.ncbi.nlm.nih.gov/34808024/>) have shown higher MYB47 expression to be associated with elevated drought survival. These papers should be cited in introduction or discussion, given for example that a link was established between glucosinolate levels and drought tolerance (<https://pubmed.ncbi.nlm.nih.gov/31492889/>).

2. The genetic material is not well defined. If this is the first description, alleles used must be precised, in particular relative to previous publications, and expression knock-out must be documented.

3. Throughout the ms, there is a confusion in the use of the term 'downregulated'. Authors conducted a RNA-seq experiment and expressed data as WT-JA FC vs WT-mock. For mutant-WT comparison, they use direct genotype comparisons of similar treatment, generating negative log2 FC. Authors then qualify this behavior as 'downregulation'. This is incorrect to me, as there is no biological downregulation from a pre-existing expression for these genes. My understanding is that in their study, authors exclusively refer to genes that are JA-upregulated in WT, and they then observe that some subgroups of genes are no longer upregulated in either myc234 or myb47_95 mutants. This should be described as 'non-responsive' or 'not induced', but implies that data showing log2 FC JA vs mock is made accessible to reader for each genotype. This brings me to another aspect of that question: authors focus on the set of genes that are JA up-regulated in a MYB47/95 dependent manner, which are probably the most abundant; but what about genes that are downregulated by JA in WT and may remain stable in myb47_95? These TFs may also shut-down a biological process. It would be useful to comment on this, at least to

mention if such a subset exists or not.

4. The experiment using mutants in MYB28/29 and MYB34/51/122 genes should be supported by a quantification of aliphatic and indolic glucosinolates to validate the conclusions.

5. Overall, experiments should be described more precisely, namely in several Figure legends, as to time of harvest for RNAseq, antibodies used for immunodetection etc...that are currently missing.

Minor concerns :

- Fig. S2 has no statistics
- L173-174: better define how box1&2 have been identified/selected
- L197: required rather than responsible
- L206: unilaterally: could a better interpretation be formulated ? MYC234 is epistatic to MYB47/95 ?
- Legend Fig. 4: 'atypical' Abolished. Correct
- L211 : timing of treatment ? should be indicated in transcriptome exp.
- L211-215: significance of the quantitative differences cannot be derived from the MDS plot. The comments should be on the relative distribution of groups.
- L217-218: it is probably more relevant to mention that out of 944_up-WT, 515 were no longer up in myc234, because these are not downregulated by JA treatment. Same for MYB47/95-dep genes. It would be useful to provide the actual Fold change JA vs mock for all genes in the 3 genotypes.
- L228: non-responsive rather than downregulated ?
- What about genes that are downregulated by JA in WT ? are some of those MYB47/95 dependent ?
- L242: there is no down-regulation in response to JA, rather a non-induction ?
- L242-253 : it is quite obvious that MYC234 are master regulators of almost the entire JA response, and MYB47/95 only control a small subset of those. Most MYB47/95-dep are also MYC234-dep. The example of VSP2 could be given earlier.
- L260: regulate rather than induce
- Fig. 6 : mention WB and antibody !
- L264: check the conclusion
- L275 : it even seems that many genes are higher in myb 47 94 JA ! This should be discussed. Is this a kind of compensation of glucosinolate biosynthetic genes balancing the defective expression of ER body structural genes ?
- L288: mediate, rather than are responsible for. This experiment should be validated by showing the impairment of glucosinolate synthesis.
- L298: this result could be shown earlier, at beginning of story. However, the pattern is not very different from free GFP, with a visible signal remaining visible in cytoplasm.
- L363: identify rather than show
- What is considered as seedling ? when in dvpt are ER bodies no longer visible ?
- L392: suggest rather than indicate
- L394: proteins rather than genes
- L406-409: does this mean this binding is not specific to MYB47/95, and that other MYBs can bind box2 ?
- L415-417: the causal connection between these two statements is not demonstrated
- L433: downregulation is not the appropriate term, as explained previously. There is no biological downregulation. It is only when comparing mutant expression to WT that negative numbers show-up. Please reformulate.
- L439: please be more accurate: most JA response genes do not depend...
- L440-41: MYB47/95 control a sub-branch of the MYC2/3/4 targets
- L499: 1 g FW + 200 µL buffer seems unrealistic. Please check

Reviewer #2

(Remarks to the Author)

Bizan and co-authors show that MYB47 and MYB95, a MYB subfamily unique to the Brassicales order, are responsible for the formation of jasmonate-inducible ER bodies. The loss of ER bodies and TSA1 and BGLU18 expressions in the myb47 myb95 knockout mutants under JA treatment supports the idea that these MYBs play a key role in the activation of ER body formation via these genes. It is also interesting that unlike MYC2, MYC3, and MYC4, MYB47 and MYB95 regulate only a subset of JA-responsive genes, including ER-body and defense-related genes. Phylogenetic analysis addresses how this specific MYB subfamily have co-evolved with ER bodies in the Brassicales order.

The story is logical and well supported by massive amounts of data to expand the insights on MYB transcriptional factors involved in ER body formation and JA-inducible stress response. The reviewer agrees this manuscript is highly likely to satisfy the broad readership of Communications Biology.

The reviewer only wonders about the substrate of BGLU18. Based on the decrease of indolic glucosinolates under JA treatment, it seems that glucosinolate breakdown is activated as a defense response. Can the authors detect that JA treatment increases glucosinolate breakdown products such as indol-3-ylmethylamine and raphanusamic acid on the LC-MS data? If BGLU18 also hydrolyzes glucosinolates, why glucosinolate levels are not affected regardless of the presence of JA-inducible ER bodies? Does myrosinase activity in the wild-type leaves change before and after JA treatment? How about the myb47 myb95 mutant? I wonder whether the physiological function of JA-inducible ER bodies is truly relevant to glucosinolates.

Minor comment:
Fig S7: The color key is missing.

Reviewer #3

(Remarks to the Author)

The manuscript by Bizan et al. presents a novel gene regulatory pathway linking jasmonate signaling to ER-body formation through two MYB transcription factors. The authors have conducted comprehensive biomolecular and genetic experiments from multiple angles to support their findings, all of which appear convincing. The manuscript is well-written and supported by a diverse range of references. Therefore, I recommend this manuscript for publication in Communications Biology. Below are suggestions to enhance the manuscript's clarity and impact.

Major comments:

1. Structure and organization: Several sections in the latter part of the Results appear somewhat disconnected from the main narrative, specifically the chapters on "Nuclear Localization of MYB47 and MYB95," "MYB47 and MYB95 Interact with MYC2, MYC3 and MYC4," and "MYB47 and MYB95 Form Unique Clade in the Phylogenetic Analysis." While these sections contain valuable information, their current placement disrupts the flow of the central story about MYB47/95 regulating jasmonate-inducible ER-body formation. Reorganizing these sections would maintain a more focused narrative on the core discovery. This would allow readers to follow the central story without getting sidetracked by supportive but less critical information.
2. Phylogenetic analysis: The phylogenetic study presented does not appear to offer substantial novelty beyond what has been previously reported in Stefanik et al. 2020 (PCP). Consider either highlighting key new insights from this analysis or integrating it more concisely into the broader narrative.
3. Discrepancies in promoter binding mechanisms: The discrepancy between the EMSA results and promoter activity analyses requires more thorough discussion. The current explanation for how the two MYBs could modulate promoter activity via box1 without direct physical interaction is not sufficiently convincing. The previous study by Stefanik et al. 2020 (PCP) suggested a possible association with MYCs rather than MYBs, which should be referenced and discussed in this context.

Version 1:

Reviewer comments:

Reviewer #1

(Remarks to the Author)

The authors have introduced modifications that answer most of my concerns. However, the following were not properly considered.

Major points :

2. The knock-out nature of the newly introduced mutant lines (myb47-2 and myb95) is still not documented, monitoring of transcript accumulation would be required to validate these lines.
3. 'reducible' may be replaced by 'repressed'
4. 'prevents' rather than 'stops'

Minor concerns :

Fig S2 statistics : it is not clear on which data the statistical test was run. On summed data ? It would be more relevant to score each category.

Reviewer #2

(Remarks to the Author)

My concerns have mostly been addressed.

Reviewer #3

(Remarks to the Author)

I am satisfied with the authors' responses to my major concerns and find the revised manuscript acceptable for publication. The authors have made meaningful improvements that address the core issues I raised.

While not required for acceptance, the authors might consider expanding the mechanistic discussion to include potential explanations for how MYB47/95 could indirectly regulate TSA1 promoter activity through protein-protein interactions,

chromatin modifications, or co-regulatory networks, though the current acknowledgment is sufficient.

Version 2:

Reviewer comments:

Reviewer #1

(Remarks to the Author)

The authors have properly addressed my remaining concerns.

Response to the reviewer's comments

Reviewer #1 (Remarks to the Author):

Major concerns:

1. Although little is known on these genes/proteins, two previous papers have described MYB47 involvement in two different biological contexts: Renard et al (<https://pubmed.ncbi.nlm.nih.gov/32519347/>) have shown MYB47 to be required for seed longevity; Marquis et al (<https://pubmed.ncbi.nlm.nih.gov/34808024/>) have shown higher MYB47 expression to be associated with elevated drought survival. These papers should be cited in introduction or discussion, given for example that a link was established between glucosinolate levels and drought tolerance (<https://pubmed.ncbi.nlm.nih.gov/31492889/>).

We cited the suggested references with an explanation in the introduction section, as follows, “MYB47 has been shown to be involved in seed longevity and drought responses^{34,35}, suggesting that this family functions during a stress response. However, the available information about the function of the MYB47/95 subfamily is limited.” (Page 4, lines 90-92)

2. The genetic material is not well defined. If this is the first description, alleles used must be precised, in particular relative to previous publications, and expression knock-out must be documented.

We modified the Plant Materials subsection to describe plant materials more precisely. We make clear the alleles of mutants used in previous papers and our paper. The modified sentences are as follows,

“T-DNA insertion mutants, *myb47-2* (SALK_123009C), *myb95-1* (GK-314F05), and *myb47-2 myb95-1* (DUPLO_2725, cross of SALK_123009C and GK-314F05)⁶⁷ were obtained from the Nottingham Arabidopsis Stock Centre (NASC). The *myb47-2* mutant is another allele of the *myb47* mutant used by Renard et al.³⁵. Homozygous plants were obtained and confirmed by PCR. The GFP-HDEL gene was introduced into these mutants by Agrobacterium-mediated transformation to visualize ER bodies¹⁰. Previously reported *myc2 myc3 myc4*⁶⁸ (a kind gift from H. Frerigmann), *myb28 myb29*²⁸ (a kind gift from B. A. Halkier), *myb34 myb51 myb122*⁶⁸ (a kind gift from H. Frerigmann), *nail*⁹ (a kind gift from I. Hara-Nishimura, also available from NASC, N69075), and *bglu18* (SALK_075731)¹⁹ were used. For the sake of simplicity, the lines are named *myb47*, *myb95*, *myb47,95*, *myc2,3,4*, *myb28,29*, and *myb34,51,122*.” (Page 19, lines 493-503)

3. Throughout the ms, there is a confusion in the use of the term ‘downregulated’. Authors conducted a RNA-seq experiment and expressed data as WT-JA FC vs WT-mock. For mutant-WT comparison, they use direct genotype comparisons of similar treatment, generating negative log₂ FC. Authors then qualify this behavior as ‘downregulation’. This is incorrect to me, as there is no biological downregulation from a pre-existing expression for these genes. My understanding is that in their study, authors exclusively refer to genes that are JA-upregulated in WT, and they then observe that some subgroups of genes are no longer upregulated in either *myc234* or *myb47_95* mutants. This should be described as ‘non-responsive’ or ‘not induced’, but implies that data showing log₂ FC JA vs mock is made accessible to reader for each genotype. This brings me to another aspect of that question: authors focus on the set of genes that are JA up-regulated in a MYB47/95 dependent manner, which are probably the most abundant; but what about genes that are downregulated by JA in WT and may remain stable in *myb47_95*? These TFs may also shut-down a biological process. It would be useful to comment on this, at least to mention if such a subset exists or not.

We agreed with the misuse of the word “downregulation” as the reviewer pointed out and replaced the words. In the current manuscript, we have used different words to distinguish true JA-downregulate genes and genes reducing JA-upregulation. Showing log₂ FC JA vs mock for each genotype will be intuitive, and we added such data in the supplemental materials. According to the comment, we reanalysed the RNAseq data and found that the genes downregulated by MYB47/95 were not many. We added the information in the text, “In contrast to JA-inducible genes, the number of JA-reducible genes controlled by MYB47/95 was small (Supplemental Table S6 and Supplemental Figure S7).” (Page 10, lines 268-269)

4. The experiment using mutants in MYB28/29 and MYB34/51/122 genes should be supported by a quantification of aliphatic and indolic glucosinolates to validate the conclusions.

These mutants originated from different research groups, and their glucosinolate levels have been evaluated already (Sønderby et al., <https://doi.org/10.1371/journal.pone.0001322>; Li et al., <https://doi.org/10.1093/pcp/pct085>; Frerigmann et al., <https://doi.org/10.1093/mp/ssu004>; Frerigmann et al., <https://doi.org/10.1016/j.molp.2016.01.006>). We have confirmed that these mutants lack aliphatic or indolic glucosinolates, respectively, but the results are confirmatory. Therefore, we referred to original research instead of showing our data and inserted the following sentence, “and the loss of these transcription factors stops the accumulation of corresponding glucosinolates^{28,29,30,32,33,46}” (Page 12, lines 314-315)

5. Overall, experiments should be described more precisely, namely in several Figure legends, as to time of harvest for RNAseq, antibodies used for immunodetection etc...that are currently missing.

We added the information about the time of harvest, antibodies, etc, in Figure legends (Figure 1C, 4A, 5A, 6B, 7).

Minor concerns :

- Fig. S2 has no statistics

We added statistical analysis.

- L173-174: better define how box1&2 have been identified/selected

We added explanations; "We have identified two conserved sequences in the *TSA1* promoter regions of five Brassicaceae plant species and named these as potential *cis*-elements, box1 (CTTGTTT[A/G]) and box2 (GTTAGTT)²⁰. The Arabidopsis *TSA1* promoter region, including box1 and box2, was used as the biotinylated DNA probe (hereafter referred to as pTSA1) (Figure 3A)." (Page 7, lines 183-187).

- L197: required rather than responsible

We replaced it.

- L206: unilaterally: could a better interpretation be formulated ? MYC234 is epistatic to MYB47/95 ?

Thanks for the suggestion. We modified the sentence, "These findings indicate that MYC2/3/4 are upstream transcription factors that regulate MYB47/95 expression in response to JA.." (Page 9, lines 218-219)

- Legend Fig. 4: 'atypical' Abolished. Correct

We changed it.

- L211 : timing of treatment ? should be indicated in transcriptome exp.

We added the information – 2 days.

- L211-215: significance of the quantitative differences cannot be derived from the MDS plot. The comments should be on the relative distribution of groups.

We modified the corresponding sentences, "In the multidimensional scaling (MDS) plot, the distribution of JA-treated *myc2,3,4* mutant is close to the mock-treated plants, suggesting a reduction of the JA response in *myc2,3,4* mutant (Figure 5A). By contrast, the distributions of *myb47,95* double mutant and wild-type showed that these plants have substantial JA responses. However, the distributions of JA-treated *myb47,95* and wild-type differ from each other in the MDS plot, suggesting that the JA response in the *myb47,95* double mutant is distinct from that in wild-type (Figure 5A)." (Page 9, lines 233-239)

- L217-218: it is probably more relevant to mention that out of 944_{up}-WT, 515 were no longer up in *myc234*, because these are not downregulated by JA treatment. Same for MYB47/95-dep genes. It would be useful to provide the actual Fold change JA vs mock for all genes in the 3 genotypes.

We rephrased the sentence. We provided the fold change data of mock vs JA-treatment within the same genotype for all genes and JA-upregulated genes as Supplementary Materials.

- L228: non-responsive rather than downregulated ?

We changed it. (Page 10, line 254)

- What about genes that are downregulated by JA in WT ? are some of those MYB47/95 dependent ?

Yes. We reanalysed the data, but the number of downregulated genes was small. As mentioned above, we added this information to the text.

- L242: there is no down-regulation in response to JA, rather a non-induction ?

We changed it. (Page 9, line 222)

- L242-253 : it is quite obvious that MYC234 are master regulators of almost the entire JA response, and MYB47/95 only control a small subset of those. Most MYB47/95-dep are also MYC234-dep. The example of VSP2 could be given earlier.

According to the comment, we moved the corresponding paragraph to the first part of the subsection (Page 9, lines 222-230).

- L260: regulate rather than induce
We changed it to “form the ER bodies”. (Page 11, line 276)
- Fig. 6 : mention WB and antibody !
We added the information in the Figure legend.
- L264: check the conclusion
We removed the unrelated description. (Page 11, line 281)
- L275 : it even seems that many genes are higher in myb 47 94 JA ! This should be discussed. Is this a kind of compensation of glucosinolate biosynthetic genes balancing the defective expression of ER body structural genes ?
We included this as “The expression of certain genes was higher in the *myb47,95* double mutant compared to the wild-type, potentially compensating for the loss of the defense mechanisms regulated by MYB47 and MYB95.” (Page 11, lines 292-294)
- L288: mediate, rather than are responsible for. This experiment should be validated by showing the impairment of glucosinolate synthesis.
We changed it. As mentioned above, the absence of glucosinolates has already been reported in these mutants, and we referred to the corresponding papers.
- L298: this result could be shown earlier, at beginning of story. However, the pattern is not very different from free GFP, with a visible signal remaining visible in cytoplasm.
We reorganized the section according to your and Reviewer 3’s comments. We deleted the section and combined it with the section of “MYB47 and MYB95 Interact with MYC2, MYC3, and MYC4”. We added images with a lookup table to highlight the difference in fluorescence signal between MYB47/95-GFP and free GFP (Supplemental Figure S11).
- L363: identify rather than show
We changed it. (Page 14, line 373)
- What is considered as seedling ? when in dvpt are ER bodies no longer visible ?
In the revised manuscript, we defined the date as 5- to 10-day-old seedlings. (Page 15, lines 385-386) The epidermis of cotyledons uniformly accumulates ER bodies, but after 11 days, ER bodies gradually disappear in the basal part of cotyledons (Matsushima et al., <https://doi.org/10.1104/pp.009464>).
- L392: suggest rather than indicate
We changed it. (Page 15, line 402)
- L394: proteins rather than genes
We changed it. (Page 15, line 404)
- L406-409: does this mean this binding is not specific to MYB47/95, and that other MYBs can bind box2 ?
Although this is possible, we do not have any experimental evidence that other MYBs can bind box2. Furthermore, we noticed that Pontier et al. (<https://doi.org/10.1046/j.1365-313x.2001.01049.x>) nor Chakravarthy et al. (<https://doi.org/10.1105/tpc.017574>) did not show direct evidence that the sequence is bound by MYBs. Therefore, this is an open question of how many MYBs recognize box2 in Arabidopsis. Because of these uncertainties, we would like to avoid deep discussion on this, but rather describe the fact that box2 is identical to the potential MYB binding sites identified in tobacco and tomato in the current manuscript, and we believe this information will be enough to support that MYB47/95 bind box2, which is our main claim in the context. However, we would like to keep this interesting question for further analysis in the future. Thank you for the comment.
- L415-417: the causal connection between these two statements is not demonstrated
We removed the description in the revised manuscript.
- L433: downregulation is not the appropriate term, as explained previously. There is no biological downregulation. It is only when comparing mutant expression to WT that negative numbers show-up. Please reformulate.

Thank you again for the comments. We rephrased it as “Transcriptome data analysis revealed reduced induction of ER body-related genes (*MEB2*, *JAL5*, *JAL6*, *JAL35*, and *At3g28220*) and defense-related genes in both the JA-treated *myc2,3,4* mutant and *myb47,95* double mutant.” (Page 17, lines 441-444)

- L439: please be more accurate: most JA response genes do not depend...
We modified it to “induction of many JA response genes does not depend” (Page 17, lines 447-448)
- L440-41: MYB47/95 control a sub-branch of the MYC2/3/4 targets
We modified it to “a subset of the MYC2/3/4 target genes that control” (Page 17, line 449)
- L499: 1 g FW + 200 μ L buffer seems unrealistic. Please check
It should be 100 mg. Thank you. (Page 20, line 523)

Reviewer #2 (Remarks to the Author):

The reviewer only wonders about the substrate of BGLU18. Based on the decrease of indolic glucosinolates under JA treatment, it seems that glucosinolate breakdown is activated as a defense response. Can the authors detect that JA treatment increases glucosinolate breakdown products such as indol-3-ylmethylamine and raphanusamic acid on the LC-MS data? If BGLU18 also hydrolyzes glucosinolates, why glucosinolate levels are not affected regardless of the presence of JA-inducible ER bodies? Does myrosinase activity in the wild-type leaves change before and after JA treatment? How about the *myb47 myb95* mutant? I wonder whether the physiological function of JA-inducible ER bodies is truly relevant to glucosinolates.

We thank the reviewer for taking the time to review our manuscript. We speculate that one of the BGLU18 substrates is glucosinolates since its close homologue, PYK10/BGLU23, has myrosinase activity against indolic and aliphatic glucosinolates (Nakano et al., <https://doi.org/10.1111/tpj.13377>; Yamada et al., <https://doi.org/10.1038/s42003-019-0739-1>). Because Arabidopsis leaves have typical myrosinase (TGG1 and TGG2), it was difficult to completely separate the myrosinase activity of BGLU18 from other myrosinases in Arabidopsis rosette leaves. However, we noticed that the indolic glucosinolate breakdown was decreased in the leaf extract of *bglu18* or *myb47,95* mutants compared with wild-type (Supplemental Figure S9), suggesting that BGLU18 has myrosinase activity.

Because BGLU18 in ER bodies cannot contact glucosinolates in the vacuole in non-homogenized leaves, the glucosinolate levels are not affected regardless of the presence of ER bodies. However, once tissues are homogenized, the reaction starts. Thus, we speculate that the observed difference in glucosinolate accumulation levels between JA-treated and non-treated intact leaves is due to an aspect different from ER body accumulation or BGLU18 activity.

There are already reports showing that JA-treatment enhances the myrosinase activities in Arabidopsis, in which they mentioned that TGG1 and TGG2 are the main contributors (Feng et al., <https://doi.org/10.1093/plphys/kiab283>; Capella et al., <https://doi.org/10.1007/s004250100548>). Because of the difficulty of separating myrosinase activities, we did not conduct the experiment of measuring myrosinase activity in the JA-treated leaves; the experiment of measuring BGLU18 myrosinase activity may fail due to high TGG1/2 activity. The question could be solved by developing more specialized materials or techniques, e.g., *tgg1 tgg2 bglu18* mutants.

There are reports that BGLU18 can hydrolyse ABA-glucoside and is involved in the drought response. We have checked stomatal opening, a hallmark of ABA response, but there seems to be no difference between wild type and *myb47,95* mutant. Since we could not find any relationship between BGLU18 and ABA-glucoside in our experimental systems, we did not focus on this direction, but rather on plant chemical defence. Therefore, we mentioned the relationship between BGLU18 and ABA-glucoside in the manuscript.

We modified the manuscript by adding a new result (Supplemental Figure S9) and inserting the following sentences in the results section,

“Because BGLU18 is a close homologue of BGLU23 that has a myrosinase activity^{5,6}, we examined whether the glucosinolate breakdown differs between wild-type and *bglu18* or *myb47,95* mutants. We prepared JA-treated rosette leaves to induce BGLU18 accumulation, then homogenized them to promote the enzyme reaction using endogenous glucosinolate substrates. We found that the breakdown of indole-3-ylmethyl glucosinolate (I3M) and 4-methoxyindol-3-ylmethyl glucosinolate (4MOI3M) was reduced in both *bglu18* and *myb47,95* mutants (Supplemental Figure S9), despite the presence of other major myrosinases known as TGG1 and TGG2^{44,45}. These findings suggest that BGLU18 possesses a myrosinase activity.” (Page 12, lines 304-312)

and the discussion section,

“Due to the high sequence similarity between BGLU18 and BGLU23, we speculated that glucosinolates might be substrates of BGLU18. Indeed, we found that I3M and 4MOI3M degradation is partly dependent on BGLU18 in the homogenate of JA-treated leaves. Because ER bodies uniquely appeared in proximity to the wounded sites, it can be concluded that wound-induced ER bodies are involved in defence against microorganism infection or herbivore attack at the wounded sites. Besides this, it has been reported that BGLU18 hydrolyses ABA-glucoside to activate ABA signaling in petioles in non-wounded leaves^{61,62}. We did not observe any ABA-related phenotype in our experiments, but it seems that BGLU18 has a dual function in Arabidopsis; one in the activation of ABA-signaling and another in chemical defense.” (Page 17-18, lines 455-464)

Minor comment:

Fig S7: The color key is missing.

We added the color key.

Reviewer #3 (Remarks to the Author):

Major comments:

1. Structure and organization: Several sections in the latter part of the Results appear somewhat disconnected from the main narrative, specifically the chapters on "Nuclear Localization of MYB47 and MYB95," "MYB47 and MYB95 Interact with MYC2, MYC3 and MYC4," and "MYB47 and MYB95 Form Unique Clade in the Phylogenetic Analysis." While these sections contain valuable information, their current placement disrupts the flow of the central story about MYB47/95 regulating jasmonate-inducible ER-body formation. Reorganizing these sections would maintain a more focused narrative on the core discovery. This would allow readers to follow the central story without getting sidetracked by supportive but less critical information.

According to the comment, we reorganized these sections. We reduced chapters and text volumes by removing non-essential explanations, and we focused more on the core discovery. (Page 13-14, lines 325-369)

2. Phylogenetic analysis: The phylogenetic study presented does not appear to offer substantial novelty beyond what has been previously reported in Stefanik et al. 2020 (PCP). Consider either highlighting key new insights from this analysis or integrating it more concisely into the broader narrative.

In Stefanik et al. (<https://doi.org/10.1093/pcp/pcz236>), we reported that the diversification of ER body-forming genes, namely *NAI2* and *TSA1*, occurs only in the Brassicaceae family. In the current manuscript, we found that *MYB47/95* (which controls *TSA1* but not *NAI2*) diversification seems to occur in the appearance of Brassicales, which means the *MYB47/95* evolution occurred before establishment of the Brassicaceae. Therefore, the novelty is that the *MYB47/95* evolved before the diversification of ER body formation. We included this information in the manuscript as follows,

“These findings suggest that the divergence of the *MYB47/95*, *MYB28/29/76*, and *MYB34/51/122* subfamilies occurred early during the evolution of the Brassicales plants, in contrast to the ER body formation, which diversified exclusively in the Brassicaceae family²⁰.” (Page 14, lines 366-369)

3. Discrepancies in promoter binding mechanisms: The discrepancy between the EMSA results and promoter activity analyses requires more thorough discussion. The current explanation for how the two MYBs could modulate promoter activity via box1 without direct physical interaction is not sufficiently convincing. The previous study by Stefanik et al. 2020 (PCP) suggested a possible association with MYCs rather than MYBs, which should be referenced and discussed in this context.

We included the suggested information with the reference (Stefanik et al., 2020) in the manuscript, as follows “This suggests that binding of another transcription factor, possibly a bHLH protein, to box1 is required for the *TSA1* promoter, since box1 is proposed as a bHLH binding site²⁰.” (Page 8, lines 200-202).

Response to the reviewer's comments

Reviewer #1 (Remarks to the Author):

Major points :

1. The knock-out nature of the newly introduced mutant lines (*myb47-2* and *myb95*) is still not documented, monitoring of transcript accumulation would be required to validate these lines.

According to the suggestion, we examined the expression levels of *MYB47* and *MYB95* in the T-DNA insertion mutant. We validated that no *MYB95* mRNA was produced in *myb95-1*, indicating it is a gene knockout mutant. Recently, the same line as *myb47-2* (SALK_123009) has been reported as a *MYB47* gene knockout line (Cao et al. *New Phytologist* (2025) 246: 2192–2206, DOI:10.1111/nph.70133). We carefully reevaluated the line and found that it has a T-DNA insertion occurring 4 bp inside the intron, retaining the ability to produce *MYB47* mRNA, but its expression was reduced. We conclude that the SALK_123009 line is a *MYB47* knockdown mutant that can be useful for analyzing the function of MYB47. Supporting this, We have observed the effect of the *myb47* mutation as the introduction of the *myb47-2* mutation in the *myb95-1* mutant enhances mutant phenotypes (Figures 1 and 2). Cao et al. have demonstrated that the SALK_123009 line exhibits a substantial *myb47* mutant phenotype. We included our result as a Supplemental Figure S2. Based on our evaluation, we stopped using “knockout” for *myb47-2* in the manuscript.

2. ‘reducible’ may be replaced by ‘repressed’

We changed it.

3. ‘prevents’ rather than ‘stops’

We changed it.

Minor concerns :

Fig S2 statistics : it is not clear on which data the statistical test was run. On summed data? It would be more relevant to score each category.

The Mann-Whitney U-test was run on all raw data to compare two groups, WT and *myb47,95*. Each category represents the ability for ER body formation that is tightly related to the other categories. Therefore, we employed rank-based statistical analysis, including all categories, to compare the groups. We did not use a category-wise comparison because each category is not independent from others. We examined a sufficient number of biological replications ($n = 68$ for WT and $n = 43$ for *myb47,95*) that should be sufficient for the conclusion.

Reviewer #3 (Remarks to the Author):

While not required for acceptance, the authors might consider expanding the mechanistic discussion to include potential explanations for how MYB47/95 could indirectly regulate TSA1 promoter activity through protein-protein interactions, chromatin modifications, or co-regulatory networks, though the current acknowledgment is sufficient.

Thanks for the suggestion. We had carefully excluded the possibility because it may cause controversy. However, indeed, our results do not exclude the possibility of indirect regulation of MYB47/95 on the *TSA1* promoter. Therefore, we remove the expression “MYB47/95 directly regulates the TSA1 promoter” from the text.